# Methodology and applicability of the human contact burn injury model: A systematic review

**Anders Deichmann Springborg**[1]*, **Caitlin Rae Wessel**[2], **Lars Peter Kloster Andersen**[3], **Mads Utke Werner**[1,4]

**1** Department of Anesthesia, Multidisciplinary Pain Center, Pain and Respiratory Support, Neuroscience Center, Copenhagen University Hospital, Copenhagen, Denmark, **2** Department of Physiology, University of Kentucky, Lexington, KY, United States of America, **3** Department of Anaesthesia and Intensive Care, Bispebjerg and Frederiksberg Hospital, University of Copenhagen, Copenhagen, Denmark, **4** Department of Clinical Sciences, Lund University, Lund, Sweden

* andersspringborg@gmail.com

## Abstract

The contact burn injury model is an experimental contact thermode-based physiological pain model primarily applied in research of drug efficacy in humans. The employment of the contact burn injury model across studies has been inconsistent regarding essential methodological variables, challenging the validity of the model. This systematic review analyzes methodologies, outcomes, and research applications of the contact burn injury model. Based on these results, we propose an improved contact burn injury testing paradigm. A literature search was conducted (15-JUL-2020) using PubMed, EMBASE, Web of Science, and Google Scholar. Sixty-four studies were included. The contact burn injury model induced consistent levels of primary and secondary hyperalgesia. However, the analyses revealed variations in the methodology of the contact burn injury heating paradigm and the post-burn application of test stimuli. The contact burn injury model had limited testing sensitivity in demonstrating analgesic efficacy. There was a weak correlation between experimental and clinical pain intensity variables. The data analysis was limited by the methodological heterogenicity of the different studies and a high risk of bias across the studies. In conclusion, although the contact burn injury model provides robust hyperalgesia, it has limited efficacy in testing analgesic drug response. Recommendations for future use of the model are being provided, but further research is needed to improve the sensitivity of the contact burn injury method. The protocol for this review has been published in PROSPERO (ID: CRD42019133734).

## Introduction

Human experimental pain models are pivotal research tools in studying mechanisms of pain pathophysiology and pharmacodynamics of analgesics [1–64] (Fig 1). While animal models are principal in understanding the basic circuitry of nociceptive pathways as well as

**Data Availability Statement:** All relevant data are within the manuscript and its Supporting Information files.

**Funding:** The author Caitlin R. Wessel received financial support by The National Institutes of

Health (NIH) grant number NIHR01DA37621 to Bradley K. Taylor.'

**Competing interests:** The authors have declared that no competing interests exist.

fundamental drug effects, human models evaluating sensory-discriminative and cognitive-evaluative aspects of pain are essential in pre-clinical research [65–67].

Experimental inflammatory pain models mimic the sensitization of primary afferent nociceptors and spinal processing seen in clinical pain [69], thus leading to primary hyperalgesia in the inflamed tissues and secondary hyperalgesia in the circumscribed 'normal/undamaged' tissues. A number of standardized inflammatory models evoking hyperalgesia are applied in experimental pain research, including the contact burn injury (CBI) model [33], the heat/capsaicin model [70], and the ultraviolet B (UVB) model [71].

The CBI-model involves tonic heating of the skin by a contact thermode inducing a first-degree burn injury. Critical factors that govern cutaneous burn injury depth are exposure temperature and exposure time, a relationship first proposed by Moritz and Henriques in 1947 [72]. In their hallmark experimental study, porcine and human skin was exposed to a contact thermode with varying exposure temperatures and times in order to establish a time-temperature graph. The generated graph indicated the lowest exposure time at a given temperature, causing microscopically verified epidermal cell death (Fig 2). Within the range of 44°C to 51°C, an increase in heating of 1°C meant that the exposure time required for the development of epidermal cell death was reduced by 50%. At 44°C, an exposure time of 6 h was required to induce epidermal cell death; thus, the authors concluded that, below this temperature, cellular

**PRISMA 2020 flow diagram for new systematic reviews which included searches of databases, registers and other sources**

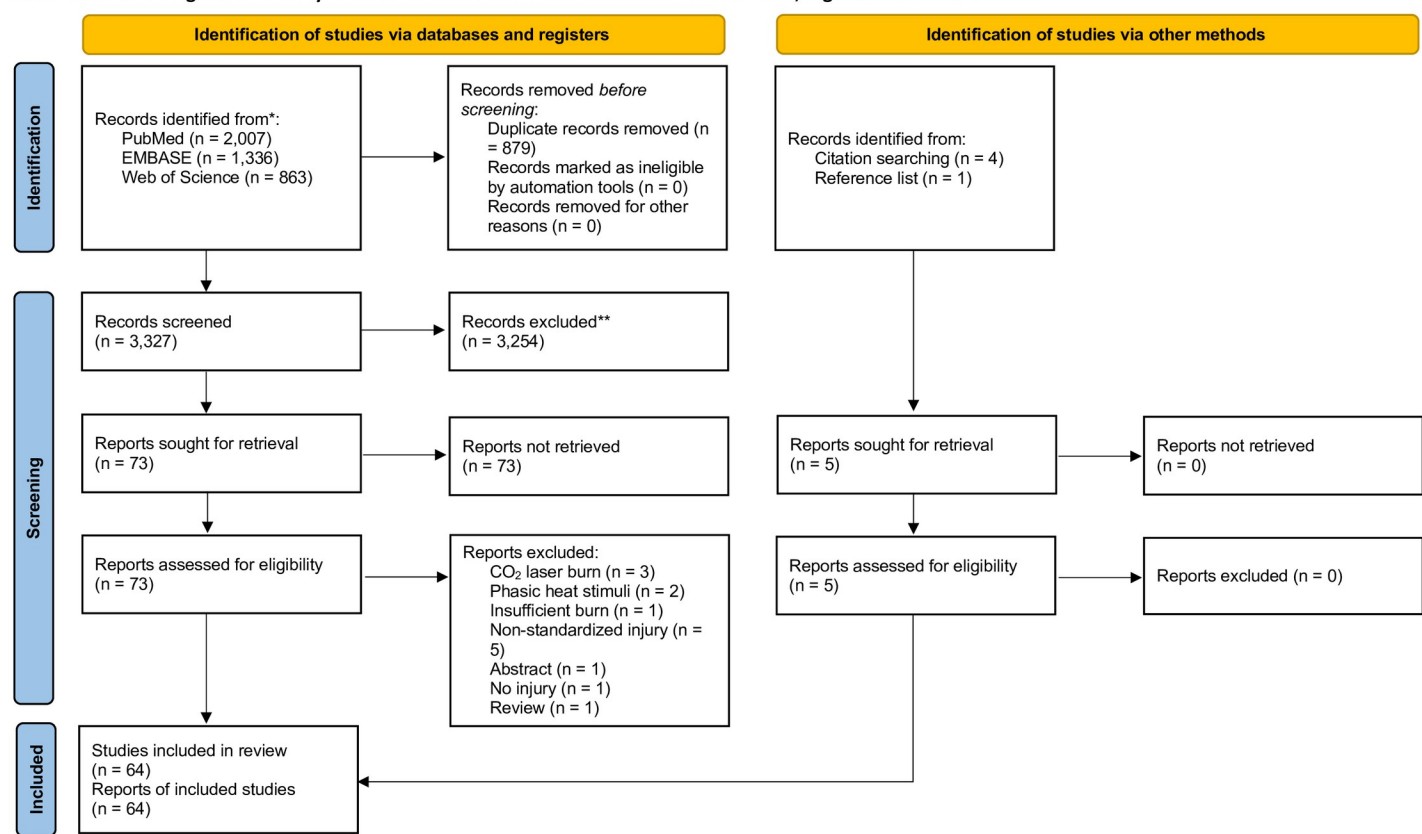

*Consider, if feasible to do so, reporting the number of records identified from each database or register searched (rather than the total number across all databases/registers).
**If automation tools were used, indicate how many records were excluded by a human and how many were excluded by automation tools.

*From:* Page MJ, McKenzie JE, Bossuyt PM, Boutron I, Hoffmann TC, Mulrow CD, et al. The PRISMA 2020 statement: an updated guideline for reporting systematic reviews. BMJ 2021;372:n71. doi: 10.1136/bmj.n71. For more information, visit: http://www.prisma-statement.org/

**Fig 1. PRISMA flow diagram for the search algorithm [68].**

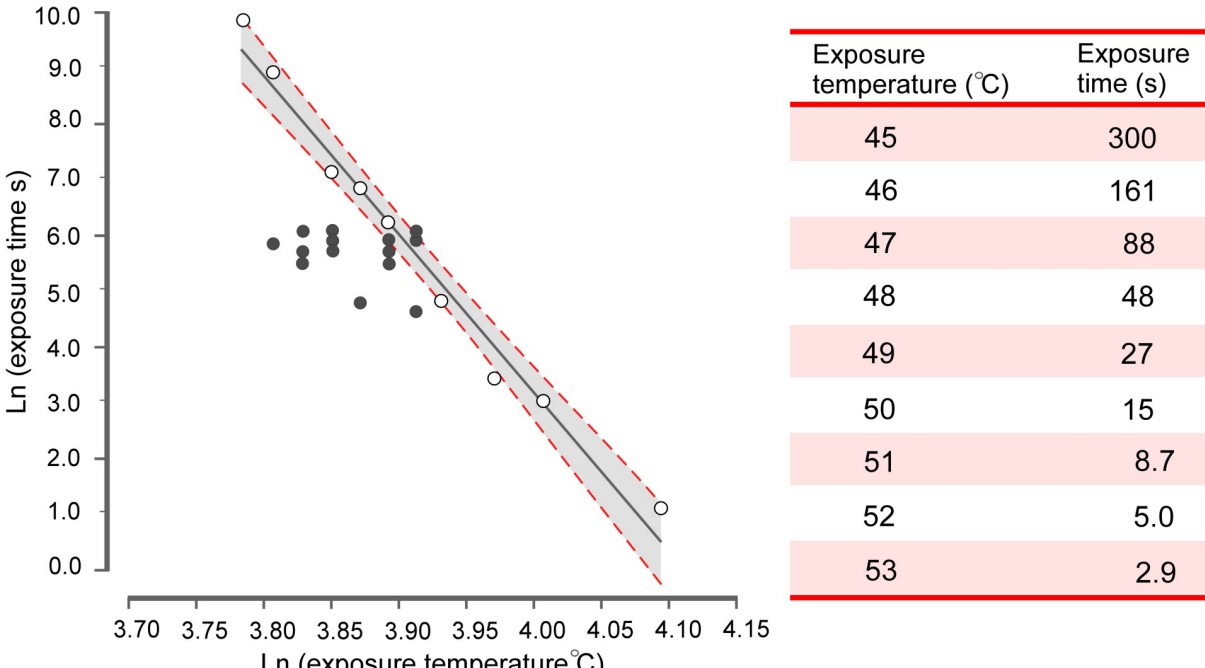

**Fig 2. Time-temperature relationship of the contact burn injury.** Double-logarithmic time-temperature relationship of the burn injury (left panel) and inclusion criteria for minimum exposure temperature and exposure time (right panel). The grey line represents a regression plot with 95% confidence bands (dashed red lines) on the logarithmic relationship between exposure temperature (X) and exposure time (Y) in producing a burn injury (modified from Naert et al. [27] based on data from Moritz and Henriques [72]; Y = -28.3*X + 116.4, $R^2$ = 0.98). The open circles represent the longest exposure time at a given temperature that failed to destroy the epidermis but induced a hyperemic skin response in humans. The red line is a parallel shift of the regression line through the heating paradigm of 300 s at 45°C (Y = -28.3*X + 113.4), serving as a 'safety margin' and inclusion criteria in this review (see right panel for specific values). The solid grey circles represent the heating paradigms of the included studies. The minimum exposure time at a given temperature that meets the inclusion criteria is outlined in the table (right panel, corresponding to the parallel shifted red line).

reparative processes were able to prevent burn injury damage [72]. Further studies have confirmed that heating the skin at 45°C for 300 s only induces transient hyperalgesia [5,70,73].

The application of the CBI-model generally involves the CBI as a conditioning stimulus with pain intensity ratings during the induction followed by Quantitative Sensory Testing, i.e., a standardized activation of the sensory system by application of graded chemical, electrical, mechanical, or thermal stimuli, with an assessment of the evoked psychophysical responses [74]. However, the employment of the CBI-model across studies generally lacks consensus regarding essential methodological variables such as exposure time, heating temperature, and contact thermode surface area. These inconsistencies present major confounding factors that may influence outcome assessments and thus impose a challenge to the general validity of the model.

The primary aim of the present study is to systematically and critically review the methodologies and outcomes of experimental human CBI-studies. The secondary aim is to propose a uniform, applicable CBI-paradigm based on a discussion of the limitations and advantages of the current studies.

## Materials and methods

A protocol is available at PROSPERO (https://www.crd.york.ac.uk/prospero/) with ID CRD42019133734. The PRISMA checklist is available as supporting information (S1 Checklist).

## Eligibility criteria

Following criteria were used to retrieve relevant articles:

1. Experimental studies applying a CBI-method in humans were considered.

2. Only studies that met the exposure temperature and exposure time criteria outlined in Fig 2 were included. The graph accommodates a parallel shift of the double-logarithmic graph original by Moritz and Henriques in order to fit the heating paradigm of 45°C for 300 s [27,72].

3. Studies applying a tonic heat stimulus (duration $\geq$ 1 min) with identical heating paradigms between subjects were included.

4. Studies applying phasic heat stimuli (stimulus duration < 1 min, or a compound burst of stimuli) were excluded.

5. Studies applying radiant heat, e.g., $CO_2$-laser, as a conditioning stimulus, were excluded.

6. Studies applying additional conditioning stimuli to the burn injury area, thereby increasing the hyperalgesic response (e.g., heat/capsaicin model), were excluded.

7. Research articles published before 1992 were not considered relevant for this review and were thus excluded.

8. Non-English studies, conference abstracts, material from textbooks, and review articles were not included.

## Search strategy and selection

A literature search was conducted in PubMed, Web of Science, and EMBASE (final search conducted 15-JUL-2020) using the following search criteria: (hyperalgesia OR pain threshold OR detection threshold OR pain sensitization OR pain measurement OR temporal summation) AND (contact heat stimulation OR heat injury OR burn injury OR local hyperthermia OR thermode) AND (healthy subjects OR healthy volunteers OR human subjects OR humans OR man OR patients). The search criteria were modified for EMBASE to fit the index terms in this database (S1 File). Google Scholar was used for forward citation chaining of all included articles after the initial reference search. The reference-lists from eligible articles were examined for additional relevant articles as well. No attempt to contact study authors for additional unpublished material was made. The authors (ADS, CRW) individually screened all identified articles based on title and abstract. In the case of disagreement between the authors regarding the relevance of an article, the final decision was made by the senior author (MUW). Subsequently, the relevant articles underwent full-text screening to determine final eligibility.

## Categorization of studies

The studies were separated into the following categories:

1. Intervention studies were characterized by the evaluation of the CBI-model in terms of both pharmacodynamic and non-pharmacodynamic interventions in reducing the hyperalgesic and/or inflammatory response of the CBI.

2. Non-intervention studies were characterized by the application of the CBI-model in all other research areas than evaluation of analgesic efficacy

The non-intervention studies were further subdivided into the following categories:

2.1. Methodological studies were characterized by the evaluation of the CBI-model in terms of validity, reproducibility, and comparison to other models of hyperalgesia.

2.2. Physiological studies were characterized by the evaluation of the CBI-model in terms of physiological mechanisms, including evaluation of imaging, pain heritability, inflammatory mediators, the contribution of the endogenous opioid system, characterization of the secondary hyperalgesia are (SHA), temporal summation, and the contribution of different receptors in the hyperalgesic response.

2.3. Predictive studies were characterized by the evaluation of the CBI-model in terms of predicting pain, primarily following surgery.

2.4. Miscellaneous studies included all CBI-studies that could not be classified into any of the previously described categories (see S2 File for data regarding these studies).

## Data extraction and data analysis

Data were extracted from relevant articles using a standardized data extraction sheet including the following information: general article information, aim, design of the study, study timeline, description of participants, description of the burn injury and additional injuries, interventions, outcome measures, post-CBI changes for each outcome, the effect of interventions on outcomes, key conclusions of the study, and study limitations. Methodological quality was assessed using the Oxford Quality Scoring System [75] for all intervention studies. This system evaluates information about randomization, blinding, and withdrawals, and drop-outs. Each study is scored on an ordinal scale of 0–5, with 0–2 representing a low-quality score and 3–5 representing a high-quality score. Before commencing data extraction for each article, the authors (ADS, CRW) individually extracted data from the same article and compared data sheets to ensure uniform and consistent data extraction. Data were synthesized into tables and analyzed qualitatively. Data are reported as median (interquartile range) unless stated otherwise.

## Results

### Literature search

A PRISMA flow diagram illustrating the number of retrieved records is presented in Fig 1. The final search resulted in 4,206 records. After the exclusion of 879 duplicates, a total of 3,327 records were analyzed. Through title and abstract review, 3,254 records were excluded. The remaining 73 records were assessed for eligibility, and 14 of these were excluded [76–89]. Five studies were additionally retrieved from reference lists and citation chaining, yielding a total of 64 studies relevant for this review [1–64]. Before commencing the literature search, the authors searched for similar reviews in the field of contact burn injuries on PubMed and PROSPERO. We found one published review from 2016 regarding the pharmacological sensitivity of the CBI model [67]. This study did not focus on methodological aspects of the CBI model in detail.

### Contact burn injury characteristics

CBI-characteristics for each study are presented in Table 1.

**Contact burn injury methodology.** Heating paradigm: The most common heating paradigm was 420 s at 47˚C (40/64 [63%] studies; Fig 2) [1,2,5,6,10,12–16,18,20,21,27,29,32–34,36–39,41–44,46,48–50,53,55–62,64]. One study applied a longer exposure time in a control group than in the intervention group (420 s vs. 360 s both at 50˚C) [7], while another study applied a

**Table 1. Contact burn injury characteristics.**

| First author | Heat paradigm | Contact area (cm²)/ manufacturer | Contact thermode pressure (kPa) | Induction site | Adverse effects | Pain rating | Sensory testing | Post-CBI changes |
|---|---|---|---|---|---|---|---|---|
| | | | | | **Intervention** | | | |
| Dahl J [7] | CTRL: 420 s at 50˚C I: 360 s at 50˚C | 3.75/Somedic | 26 | Calf | Blistering 4-8h post-injury (100%) | NR | CTRL: PI, 40′, 70′, 100′, 130′, 160′, 190′ I: PI¤, 40′, 70′, 100′, 130′, 160′, 190′† | ↑90WDT$_{in}$, ↓WDT$_{out}$††, ↓HPT$_{in}$, P-SHA*, B-SHA*, ↓Flare†††, →HPT$_{out}$ |
| Møiniche S [22] | 360 s at 49˚C | 3.75/Somedic | NR | Calf | Blistering (n NR) | NR | 24h, 72h, 168h, 14 days | ↑Flare, ↑Blistering |
| Møiniche S [25] | 300 s at 49˚C | 3.75/Somedic | NR | Calf | Blistering (n NR) | NR | PI¤, 30′¤, 60′, 120′, 180′, 240′, 300′, 360′ | ↓HPT, ↓PPT |
| Møiniche S [24] | 240 s at 49˚C | 3.75/Somedic | NR | Calf | Blistering (n NR) | NR | -90′¤, -5′, 2h, 4h, 6h | ↓HPT, ↓HPTo, P-SHA*, ↓MPT, ↑EI |
| Brennum J [5] | 420 s at 47˚C | 12.5/Somedic | 8 | Calf | No | NR | -30′¤, 150′, 330′, 390′ (+CBI at 180′) | ↓HPT, ↓HPTo, P-SHA*, B-SHA*, →WDT, →Flare, →ST |
| Møiniche S [26] | 300 s at 49˚C | 3.75/Somedic | NR | Calf | Blistering (n NR) | NR | -1.5h¤, -5′, 3h, 6h, 12h, 24h | ↓HPT, ↓HPTo, ↑P-SHA, ↓MPT, ↑EI |
| Pedersen JL [35] | 300 s at 49˚C | 3.75/Somedic | NR | Calf | Blistering (n NR) | NR | -1h¤, -5′, 3h, 24h, 48h, 72h | ↓HPT, ↓HPTo, P-SHA*, ↓MPT, ↑EI |
| Ilkjær S [13] | 420 s at 47˚C | 12.5/Somedic | 8 | Calf | No | eVAS | PI¤, -5′, 40′, 110′, 180′ | ↓HPT, P-SHA*, B-SHA* |
| Lundell JC [17] | 100 s at 50˚C | 1/MIF | NR | Forearm | No | Magnitude estimation | ~-90′¤, ~-50′, ~10′ | ↓HPT, ↑H-pain |
| Pedersen JL [30] | 300 s at 49˚C | 3.75/Somedic | 4.5 | Calf | Blistering (n NR) | NR | PI¤, 16h, 20h, 24h, 48h | HPT?, P-SHA*, MPT?, EI?, ↑Blistering |
| Pedersen JL [31] | 300 s at 49˚C | 3.75/Somedic | 4.5 | Calf | Blistering (n NR) | Verbal rank score (0–100) | PI¤, 1h, 4h, 6h, 8h, 10h, 12h | P-SHA*, MPT?, EI?, ↑Blistering |
| Warncke T [52] | 300 s at 47˚C | 12.5/Somedic | 8 | Calf | No | NR | **B$_1$**: PI¤, 90′, 150′, 210′, 270′ **B$_2$**: PI¤, -10′, 30′, 60′, 90′ | **B$_1$**: ↓HPT, ↑P-SHA, ↓HPTo **B$_2$**: ↓HPT, P-SHA*, ↓HPTo |
| Ilkjær S [12] | 420 s at 47˚C | 12.5/Somedic | 8 | Calf | No | eVAS | PI¤, -15′, 1h, 2h | ↓HPT, P-SHA*, B-SHA* |
| Pedersen JL [36] | 420 s at 47˚C | 12.5/Somedic | NR | Calf | Blistering (n NR) | VAS | PI¤, PI, 1h, 2h, 4h, 6h | ↑EI, P-SHA*, B-SHA*?, MPT?, HPT?, H-pain?, ST? |
| Petersen KL [39] | 420 s at 47˚C | 12.5/Somedic | 8 | Calf | Blistering (n = 8) | eVAS | 1h, 2h, 3h | P-SHA*, B-SHA*, B-pain$_{out}$* |
| Warncke T [53] | 420 s at 47˚C | 12.5/Somedic | 8 | Calf | NR | NR | PI¤, -10′, 30′, 90′, 150′ | ↓HPT$_{in}$, ↓MPT$_{in}$, P-SHA*, TS$_{M,out}$§§§§, →TDT$_{in}$, HPT$_{out}$?, WDT$_{in/out}$?, TDT$_{out}$?, MPT$_{out}$?, ST? |
| Warncke T [54] | 360 s at 47˚C | 12.5/Somedic | 8 | Calf | No | NR | -10′¤, 30′¤, 65′, 95′, 125′ | ↓HPT$_{in}$, ↓MPT, ↑P-SHA, ↑B-SHA, ↑TS$_{M,out}$, ↑ST, →WDT, →|CDT|, →CPT, →TDT, →HPT$_{out}$ |
| Pedersen JL [32] | 420 s at 47˚C | 12.5/Somedic | 4.5 | Calf | Blistering (20%), skin discoloration for 3 weeks (25%) | VAS | -70′¤, -40′, 0′, 60′, 120′ | ↑WDT, ↓HPT, ↑H-pain, ↑P-SHA, ↓MPT, ↑M-pain$_{in}$, ↑M-pain$_{out}$, B-SHA* |
| Hammer NA [10] | 420 s at 47˚C | 12.5/Somedic | 6.9 | Calf | No | VAS | PI¤, -15′, 0′, 1h, 2h, 3h | ↓MPT, ↑M-pain$_{in/out}$, ↑P-SHA, ↓HPT, ↑H-pain$_{short}$, B-SHA?, H-pain$_{long}$? |
| Mikkelsen S [20] | 420 s at 47˚C | 12.5/Somedic | NR | Calf | NR | NR | -5′¤, 60′¤, 90′, 135′ | ↓HPT, P-SHA*, B-SHA* |

*(Continued)*

**Table 1.** (*Continued*)

| First author | Heat paradigm | Contact area (cm²)/ manufacturer | Contact thermode pressure (kPa) | Induction site | Adverse effects | Pain rating | Sensory testing | Post-CBI changes |
|---|---|---|---|---|---|---|---|---|
| Sjölund KF [49] | 420 s at 47˚C | 12.5/Somedic | NR | Calf | Minor blistering (n = 2) | NR | -15´¤, 0´, 60´, 120´ | ↓HPT$_{in}$, ↓MPT$_{in/out}$, P-SHA*, B-SHA*, →\|CDT$_{in/out}$\|, →WDT$_{in/out}$, →HPT$_{out}$, →H-pain$_{in/out}$, →M-pain$_{in/out}$ |
| Holthusen H [11] | 300 s at 47˚C‡ | 4.5/Melcor | 4.5 | Forearm | No | eVAS | PI¤, 0´, 30´, 60´, 90´, 120´, 180´, 240´, 24h | ↓HPT, ↑P-SHA, ↓MPT, WDT?, TDT? |
| Lillesø J [16] | 420 s at 47˚C | 12.5/Somedic | 6.9 | Calf | No | VAS | PI¤, PI, 0´, 60´, 120´, 175´, 235´ | ↑WDT‡‡, ↓HPT, ↑H-pain, ↑P-SHA*, B-SHA*, ↓MPT, ↑M-pain$_{in/out}$, →\|CDT\| |
| Mikkelsen S [21] | 420 s at 47˚C | 12.5/Somedic | NR | Calf | NR | eVAS | -20´¤, 20´, 80´, 140´, 200´, 260´ | ↓HPT, P-SHA*, B-SHA* |
| Warncke T [55] | 420 s at 47˚C | 12.5/Somedic | 8 | Calf | No | NR | -30´¤, 30´, 110´, 150´ | ↓HPT$_{in}$, ↓MPT$_{in/out}$, ↑TS$_{M,out}$, P-SHA*, B-SHA*, ↑ST$_{in}$##, →ST$_{out}$, →WDT, →TDT, →HPT$_{out}$ |
| Brennum J [6] | 420 s at 47˚C | 12.5/Somedic | 2.7 | Calf | No | eVAS | -10´¤, 60´¤, 120´, 180´, 240´ | ↓HPT, ↑H-pain$_{short}$, ↑H-pain$_{long}$, P-SHA*, B-SHA* |
| Werner MU [60] | 420 s at 47˚C | 12.5/Somedic | 5.5 | Calf | Minor blistering (n = 3) | VAS | -180´¤, -5´, 0´, 60´, 120´, 180´ | ↓HPT, ↓MPT, P-SHA*, B-SHA*, ↑M-pain$_{in/out}$, ↑WDT‡‡, →\|CDT\| |
| Werner MU [58] | 420 s at 47˚C | 12.5/Somedic | 5.5 | Calf | NR | VAS | -120´¤, 60´, 120´, 180´, 240´ | ↑EI, P-SHA*, B-SHA*, ↓MPT, ↑M-pain, ↓HPT, ↑H-pain, →WDT, →\|CDT\| |
| Werner MU [59] | 420 s at 47˚C | 12.5/Somedic | 6.5 | Calf | NR | VAS | -5´¤, 55´, 85´, 125´, 165´, 205´ | ↓HPT, P-SHA*, ↓MPT, ↓M-pain$_{in/out}$, ↑EI, ↑ST$_{in}$, →\|CDT\|, →WDT, ST$_{out}$? |
| Schulte H [47] | 420 s at 46˚C | 12.5/Somedic | 1.9 | Calf | No | NR | -15´¤, 15´¤, 45´, 75´, 105´ | P-SHA*, ↓MPT$_{in/out}$, ↑TS$_{M,in}$, ↑TS$_{M,out}$ TDT$_{in/out}$? |
| Schulte H [48] | 420 s at 47˚C | 12.5/Somedic | NR | Forearm | No | NR | -55´¤, 30´¤, 155´, 275´ | ↓MPT$_{in/out}$, ↑TS$_{M,in}$, ↑TS$_{M,out}$, P-SHA* |
| Robertson L [63] | 120 s at 48˚C | 0.79/TC | 1.3 | Forearm | NR | NR | 30´¤, ~40´¤, ~50´ | ↓HPT#, ↑H-pain#, ↑M-Pain# |
| Stubhaug A [51] | 300 s at 47˚C | 12.5/Somedic | 8 | Abdomen | No | eVAS | -45´¤, 30´¤, 60´, 105´, 165´ | ↓HPT, P-SHA* |
| Ravn P [43] | 420 s at 47˚C | 12.5/Somedic | NR | Calf | NR | VAS | PI¤, 1h, 2h, 3h | →HPT, →\|CDT\|, P-SHA?, MPT?, WDT? |
| Andersen LPH [1] | 420 s at 47˚C | 12.5/Somedic | NR | Calf | NR | VAS | -90´¤, 60´, 120´, 240´, 360´ | ↓MPT$_{in/out}$, ↓HPT, ↑DT, ↑EI, ↑P-SHA, →WDT |
| Rasmussen VM [41] | 420 s at 47˚C | 12.5/Somedic | NR | Calf | No | VAS | -15´¤, 30´, 70´, 110´, 160´, 220´ | ↓HPT, ↓MPT, ↑EI, →\|CDT\|, →WDT, P-SHA? |
| Wahl AM [64] | 420 s at 47˚C | 12.5/Somedic | NR | Calf | No | VAS | -20´¤, 100´, 160´, 220´ | ↓HPT, ↓MPT, →\|CDT\|, →DT, ↑P-SHA, SE?, WDT? |
| **Non-intervention** | | | | | | | | |
| Methodological | | | | | | | | |
| Møiniche S [23] | 300 s at 49˚C | 3.75/Somedic | 26 | Calf | Blistering (n NR) | NR | PI, 3h, 6h, 24h, 30h, 48h, 54h, 72h | ↓HPT, ↓HPT$_{o}$¤¤¤¤, P-SHA*, B-SHA*, ↓MPT, ↑EI |
| Pedersen JL [33] | 420 s at 47˚C | 12.5/Somedic | 4.5 | Calf | Blistering (20%), skin discoloration for 3 weeks (25%) | VAS | -105´, -30´, 0´, 1h, 2h, 4h, 6h | ↑WDT‡‡, ↓HPT, ↑H-pain, ↑\|CDT\|‡‡‡, ↑P-SHA, B-SHA*, ↓MPT, ↑EI |
| Yucel A [61] | 420 s at 47˚C | 12.5/Somedic | 4.5 | Forearm | NR | eVAS | ~-30´, 60´, ~70´, ~80´ | P-SHA*, B-SHA*, SBF$_{in/out}$?, ST?, ↑Flare |

(*Continued*)

**Table 1.** (Continued)

| First author | Heat paradigm | Contact area (cm²)/ manufacturer | Contact thermode pressure (kPa) | Induction site | Adverse effects | Pain rating | Sensory testing | Post-CBI changes |
|---|---|---|---|---|---|---|---|---|
| Yucel A [62] | 420 s at 47°C | 12.5/NR | 4.5 | Forearm | No | eVAS | -5′, 60′ | ↓HPT-SS$_{out}$, ↓HPT-RS$_{out}$, P-SHA*, B-SHA*, ↑MI-Pain$_{in}$, ↑M-Pain$_{in/out}$, →TS$_{H,out}$, →TS$_{I,in}$, →MI-Pain$_{out}$, ↓EPT-SS$_{out}$, ↓EPT-RS$_{out}$, TS$_{E,out}$?, ↑TS$_{I,out}$, ↑TS$_{M,in/out}$, ↑Flare |
| Naert ALG [27] | 420 s at 47°C | 9/Medoc | NR | Thigh | NR | eVAS | Only QST before CBI§§§ | NA |
| Bishop T [4] | 330 s at 45°C | 10.24/Medoc | 2.9 | Forearm | No | NR | PI, 15′ | ↑P-SHA, ↑B-SHA, ↓MPT, ↑Flare, ↑SBF, →HPT |
| Ringsted TK [44] | 420 s at 47°C | 12.5/Somedic | NR | Calf | NR | VAS | 45–75′¶ | P-SHA* |
| | | | | | Physiological | | | |
| Pedersen JL [29] | 420 s at 47°C | 12.5/Somedic | 4.5 | Calf | Blistering (n NR) | VAS | PI, 0′, 1h, 2h, 4h | P-SHA*, ↓MPT, ↓EPT-SS$_{in}$, →EPR$_{in/out}$, →TS$_{E,in/out}$, →EPT-SS$_{out}$ |
| Pedersen JL [34] | 420 s at 47°C | 12.5/Somedic | 4.5 | Calf | Blistering (20%), skin discoloration for 3 weeks (25%) | VAS | -70′, -40′, 0′, 1h, 2h | ↓HPT, ↓H-pain$_{in/out}$, ↑P-SHA, ↓MPT, ↑M-pain$_{in/out}$, B-SHA*? |
| Schulte H [46] B | 420 s at 47°C | 12.5/Somedic | 1.9 | Calf | Minor blistering (n = 3) | NR | -10′, 5′, 15′, 30′, 45′, 60′ | P-SHA*, ↑Flare, ↑ST$_{in}$§, ↑ST$_{out}$§§, ↑SBF$_{area}$ |
| Norbury TA [28] | 330 s at 45°C | 0.3/Medoc | 2.7 | Forearm | NR | NRS | PI, 15′ | ↓HPT, P-SHA*, B-SHA*, ↑Flare |
| Drummond PD [8] | 120 s at 48°C | 3.1/Custom-built | NR | Forearm | NR | NRS | PI, 30′¶¶ | ↓HPT, ↑H-pain, EPS* |
| Robertson L [45] | 120 s at 48°C | 0.8/TC | 1.3 | Hand | NR | NR | 30′, ~35′, ~45′, ~50′ | ↓HPT#, ↑ST# |
| Drummond PD [9] | 120 s at 48°C | 3.1/Custom-built | NR | Forearm | No | NR | PI¤, PI, ~30′, ~60′, ~85′ | ↓HPT, ↑H-pain |
| Kupers R [15] | 420 s at 47°C | 9/Medoc | NR | Thigh | NR | eVAS | Only QST before CBI | NA |
| Petersen LJ [40] | 300 s at 49°C | 3.75/Somedic | 4.5 | Calf | NR | NR | No QST | NA |
| Kupers R [14] | 420 s at 47°C | 9/Medoc | NR | Thigh | NR | eVAS | Only QST before CBI | NA |
| Pereira MP [38] | 420 s at 47°C | 12.5/Somedic | NR | Calf | Hyperpigmentation 23 days post-injury (n = 1) | VAS | PI, 1h, 2h, 3h†††† | ↑P-SHA |
| Asghar MS and Pereira MP [2] | 420 s at 47°C | Screening: 12.5/ Somedic Experimental: 9/ Medoc | NR | Calf | NR | NRS | S-day: PI, 0′ E-day: PI, 100′ | P-SHA?, M-pain$_{in/out}$? |
| Pereira MP [37] | 420 s at 47°C | 12.5/Somedic | NR | Calf | NR | VAS | PI, 1h, 2h, 3h‡‡‡‡ | ↓HPT, ↑P-SHA, B-SHA?, WDT? |
| Slimani H [50] | 420 s at 47°C | 12.5/Somedic | NR | Calf | No | eVAS | -10′, 60′, 24h | ↓H-pain$_{AF,out}$**, ↓H-pain$_{CF,in}$###, ↓H-pain$_{CF,out}$***, ↑HPT$_{AF,in}$**, ↑HPT$_{CF,in}$####, ↑HPT$_{CF,out}$####, P-SHA*, ↓RTFD$_{AF,in}$, →H-pain$_{AF,in}$, →RTFD$_{CF,in/out}$, →RTFD$_{AF,out}$, →HPT$_{AF,out}$ |
| | | | | | Predictive | | | |
| C₁: Werner MU [57] C₂: Werner MU [56] | C₁/C₂: 420 s at 47°C | C₁/C₂: 12.5/ Somedic | C₁/C₂: 6.5 | C₁/C₂: Calf | C₁/C₂: NR | C₁/C₂: VAS | C₁: -5′, 60′¶¶¶ C₂: -5, 60′ | C₁/C₂: ↓HPT, ↑H-pain, P-SHA*, ↓MPT, ↑M-pain |
| Ravn P [42] | 420 s at 47°C | 12.5/Somedic | NR | Calf | Blistering (n = 18)¤¤ | VAS | PI, 1h, 2h, 3h | ↑WDT¤¤¤, ↓HPT, ↑P-SHA¶¶¶¶, ↓MPT$_{in/out}$, →|CDT| |
| Lunn TH [18] | 420 s at 47°C | 12.5/Somedic | NR | Thigh | No | eVAS | Only QST before CBI | NA |
| | | | | | Miscellaneous | | | |

*(Continued)*

**Table 1.** (Continued)

| First author | Heat paradigm | Contact area (cm²)/ manufacturer | Contact thermode pressure (kPa) | Induction site | Adverse effects | Pain rating | Sensory testing | Post-CBI changes |
|---|---|---|---|---|---|---|---|---|
| Matre D [19] | 300 s at 46°C | 12.5/Somedic | NR | Forearm | NR | eVAS | 0′ | P-SHA*, B-SHA*, MPT*, M-pain* |
| Aslaksen PM [3] | 240 s at 46°C | 9/Medoc | NR | Forearm | NR | VAS | No QST | NA |

**AF** = A-fibers; **B-pain** = motor brush-evoked pain ratings; **B-SHA** = brush secondary hyperalgesia area (allodynia); **CBI** = contact burn injury; **CBI-pain** = CBI-induced pain ratings; **CDT** = cool detection threshold; **CF** = C-fibers; **CTRL** = control; **DT** = dermal thickness; **EI** = erythema index; **EPR** = electrical pain response to a single stimulus; **EPS** = electrically-evoked pain sensations; **EPT** = electrical pain threshold; **EPT-RS** = electrical pain threshold repetitive stimuli; **EPT-SS** = electrical pain threshold single stimulus; **eVAS** = electronic visual analog scale; **Flare** = area of flare; **H-pain** = heat-evoked pain rating; **HPT** = heat pain threshold; **HPTo** = heat pain tolerance; **I** = intervention; **M-pain** = punctate mechanical-evoked pain rating; **Md** = median; **MI-pain** = mechanical impact-evoked pain ratings; **MIF** = Medical Instrumentation Facility of Yale University School of Medicine; **MPT** = mechanical pain threshold; **NA** = not applicable; **NR** = not reported; **P-SHA** = punctate secondary hyperalgesia area; **PI** = pre-burn injury; **PPT** = pressure pain threshold; **PPTo** = pressure pain tolerance; **QST** = quantitative sensory testing; **RTFD** = reaction time frequency distribution; **SBF** = skin blood flow; **ST** = skin temperature; **TC** = thermocouple-controlled cautery unit, purpose-built; **TDT** = tactile detection threshold; **TS_M/TS_E/TS_I/TS_H** = temporal summation to punctate mechanical/electrical/mechanical impact/heat stimulation; **VAS** = visual analog scale; **WDT** = warmth detection threshold.

* not assessed at baseline.

** significant at 24h post-CBI.

*** significant at 1h post-CBI for both sham and burn site, but only remained decreased in the burn group at 24h.

¤ pre-intervention assessment.

¤¤ due to malfunction.

¤¤¤ only significant at 1h post-CBI.

¤¤¤¤ only measured on one of the two experimental days.

# compared only to the control site.

## minor increase only on the first measurement at 30′ post-CBI on all experimental days.

### significant at 1h post-CBI, but no difference between sham and burn site.

#### significant only compared to baseline, similar for both sham and burn group.

† additional pre-intervention time-points before CBI on pre-injury block day and at 30′ post-CBI on post-injury block day.

†† only decreased on day 1 of control day, and not day 2.

††† developed in all subjects but none at 160′ post-CBI.

†††† additional assessment time-points at 72h post-CBI (pre-naloxone infusion) and 73h post-CBI (post-naloxone infusion).

‡ preliminary study with heat paradigm of 300 s at 49°C where burns resulted in second degree burns.

‡‡ significant only at 0′ post-CBI.

‡‡‡ significant only at 0′ post-CBI except in injury 2 where it was still present at 1h post-CBI.

‡‡‡‡ additional assessment time-points at 165h post-CBI (pre-naloxone infusion), and at 30′, 45′, and 60′ post-infusion.

§ back to baseline at 60′ post-CBI.

§§ back to baseline at 30′ post-CBI.

§§§ 12 subjects were retested after 3 months.

§§§§ not compared pre- to post-CBI, but temporal summation occurred in 8/10 subjects post-CBI.

ꞓ 2 observers, SHA assessments in two parts of 15 min each.

ꞓꞓ burn injury day, assessment time-points were repeated for each of 3 sites.

ꞓꞓꞓ 6 days before and 1 day after surgery.

ꞓꞓꞓꞓ no P-value reported, but 88/100 developed secondary hyperalgesia. Arrows indicate significant increases (↑) or decreases (↓), or no difference (→) post-CBI compared to pre-CBI; '?' indicates that the post-CBI change was unclear. The **Sensory testing** column contains time-points of sensory testing in relation to the burn injury induction (time 0); negative values thus signify assessments pre-CBI, whereas positive values signify assessments post-CBI (including 0′ indicating immediately after the CBI); **PI** indicates an unspecified pre-burn injury time-point.

higher temperature during a preliminary trial compared to the intervention trial (300 s at 49˚C vs. 300 s at 47˚C) [11].

Contact thermode surface area: The most commonly applied active thermode surface area were 12.5 cm$^2$ (41/64 [64%] studies) [1,5,6,10,12,13,16,18–21,29,32–34,36–39,41–44,46–62,64], 3.75 cm$^2$ (10/64 [16%] studies) [7,22–26,30,31,35,40], and 9 cm$^2$ (5/64 [8%] studies) [2,3,14,15,27]. The remaining 8/64 (13%) studies used active areas ranging from 0.3–10.2 cm$^2$ [4,8,9,11,17,28,45,63].

Induction site of CBI: Induction sites were the calf (46/64 [72%] studies) [1,2,5–7,10,12,13,16,20–26,29–44,46,47,49,50,52–60,64], volar forearm (12/64 [19%] studies) [3,4,8,9,11,17,19,28,48,61–63], thigh (4/64 [6%] studies) [14,15,18,27], abdomen (1/64 [2%] studies) [51], or dorsum of the hand (1/64 [2%] studies) [45].

Contact thermode application pressure: Applied contact thermode pressures were reported in 34/64 (53%) studies [4–7,10–13,16,23,28–34,39,40,45–47,51–55,57–63], with a median (range) of 5 (1.3 to 26) kPa.

Pain ratings: In 41/64 (64%) studies, pain ratings were reported during CBI-induction. Pain intensity assessments were: in 36 studies with the visual analog scale (VAS) [1,3,6,10–16,18,19,21,27,29,32–34,36–39,41–44,50,51,56–62,64], 15 of which were an electronic visual analog scale [6,11–15,18,19,21,27,39,50,51,61,62]; in three studies a numeric rating scale [2,8,28]; one study a verbal rank scale [31]; and in one study a 'magnitude estimation' method [17].

**Post-burn changes.** This section describes the changes induced by the CBI as a conditioning stimulus.

Primary hyperalgesia area, thermal stimuli: Warmth detection thresholds were generally not altered [1,5,41,49,54,55,58–60], but in five studies [16,32,33,42,60] increased post-CBI. In these studies, the duration of the change was less than 1 h [16,33,60]. Only one study observed a significant numerical increase in cool detection threshold following the CBI [33], while ten studies observed no post-CBI changes in cool detection threshold [16,41–43,49,54,58–60,64]. A total of 37 studies reported primary hyperalgesia to heat stimulation after the CBI, assessed as reduced heat pain threshold (HPT) or increased suprathreshold heat pain response [1,5–13,16,17,20,21,23–26,28,32–35,37,41,42,49–60,64], while two studies did not report any sensory changes [4,43]. Primary hyperalgesia lasted between 24–48 h post-CBI in one study (300 s at 49˚C, 3.75 cm$^2$) [23].

Primary hyperalgesia area, mechanical stimuli: Three studies investigated tactile detection thresholds and found no post-CBI sensory changes [53–55]. Post-CBI primary hyperalgesia to punctate mechanical stimuli was a consistent finding [1,4,10,11,16,23,24,26,29,32–35,41,42,47–49,53–60,62,64].

Secondary hyperalgesia, thermal/electrical stimuli: Secondary hyperalgesia to heat was observed in two studies [34,62], but this finding was not corroborated by four other studies [7,49,54,55]. Further, Aδ-fibers, but not C-fibers, were found to be sensitized to short radiant heat stimuli in the primary hyperalgesia area 24 h post-CBI, while no sensitization was present in the SHA [50]. Hyperalgesia to electrical stimulations was induced in the SHA in one study [62], but was not corroborated by another [29].

Secondary hyperalgesia, mechanical stimuli: A total of 15 studies [1,4,10,11,16,26,32–34,37,38,41–43,52,54,64] assessed SHAs to punctate mechanical stimulation during baseline conditions while 28 studies [5–7,12,13,20,21,23,24,28,29,35,36,44,46–51,53,55–60,62] assessed post-CBI areas without baseline assessments. In all studies the CBI induced robust SHA to punctate mechanical stimuli, with few studies reporting sustained SHA at 24 h (300 s at 49˚C, 3.75 cm$^2$) [23,26]. Areas of allodynia, during or after the CBI, were assessed in 25 studies [4,6,7,10,12,13,16,19–21,23,28,32–34,36,37,39,49,54,55,58,60–62]. Allodynia, tested by brush,

developed shortly after the start of the CBI induction, but was generally more evanescent than SHA to punctate mechanical stimuli.

Temporal summation: Studies evaluated the following stimulation modalities: electrical [29,62], thermal [62], punctate mechanical [47,48,53–55,62], and mechanical impact [62]. Temporal summation was generally demonstrated post-CBI. Further, temporal summation to thermal stimuli did not increase post-CBI [62].

Objective inflammatory variables: Erythema was measured in eleven studies and was generally long-lasting (up to 48 h) [1,23,24,26,33,35,36,41,58,59,64]. Blood flow was increased in the primary CBI-area [4,46]. Skin temperatures were increased in four studies [46,54,55,59], although, unaltered in one study [5]. Skin temperatures returned to baseline values 60 min post-CBI in the primary hyperalgesia area, while temperatures concomitantly were increased outside the CBI-area and then returned to baseline after 30 min [46]. An area of flare was reported in six studies [4,7,22,28,46,62]. Flare was not present 160 min post-CBI [7]. Dermal thickness, measured by a high-resolution ultrasound scanner, was found to increase post-CBI for up to 360 min [1,64].

Adverse effects: In 19/64 (30%) studies, blistering was reported [7,22–26,29–36,39,42,46,49,60]. In 21/64 (33%) studies the presence of blistering was examined and did not occur [5,6,9–13,16–18,38,41,47,48,50–55,62], while in 23/64 (36%) studies blistering was not examined, nor reported [1–4,8,14,15,19–21,27,28,37,40,43–45,56–59,61,63,64]. Three studies reported slight localized skin color changes in 25% of subjects persisting for three weeks [32–34], while hyperpigmentation occurred in one subject 23 days post-CBI [38].

## Study characteristics

Study characteristics are presented in Table 2.

**Intervention studies.** The 37 studies [1,5–7,10–13,16,17,20–22,24–26,30–32,35,36,39,41,43,47–49,51–55,58–60,63,64] evaluated ten drug groups (antiarrhythmic agents [49], gabapentinoids [60], glucocorticoids [35,51,58], glutamate receptor antagonists [10], local anesthetics [7,11,22,30], melatonin [1], N-methyl-D-aspartate (NMDA) receptor antagonists [12,13,20,21,32,47,53–55], non-steroidal anti-inflammatory drugs (NSAIDs) [17,24,26,39,51,52], opioids [5,16,25,43,47,48,54,55,63], and opioid antagonists [6]), along with hyperbaric oxygen [41,64], local cooling [59], and nerve blocks [31,36].

The total number of per-protocol subjects was 641, with a median of 17 (12 to 22) subjects per study. All but one study mentioned the ratio of males to females [17], and another study only described the intention-to-treat group's male to female ratio [36]. For the remaining studies, a gender ratio (males/females) of 6.4 (532/83) was present. A total of eleven studies reported the bodyweight of subjects [1,5,22,24–26,35,41,43,51,64], while four of these reported BMI [1,41,43,64]. Nine studies included an a priori sample size estimate [1,41,43,47,51,58–60,64]. Four studies [6,22,26,52] performed post hoc power-analyses, while an additional four studies estimated post hoc minimal detectable differences for all outcomes [30,32,36,60]. After an incorrect a priori sample size estimate, one study presented a post hoc power analysis [1].

All studies were described as randomized. A total of 29/37 studies were double-blind [1,6,10–13,16,17,20,21,24–26,30,32,35,39,43,47–49,51–55,58,60,63], and all were, along with two single-blinded studies [36,64], placebo-controlled. Two of the placebo-controlled studies used both active and inactive placebos [48,53]. The non-pharmacodynamic studies used control-conditions (i.e., ambient atmospheric pressure [41,64] and ambient temperature of contact thermode [59]).

Among the 34 studies involving drug administration, seven studies evaluated the effect of several drugs [22,43,47,48,51,54,55], and three studies administered naloxone along with other

**Table 2. Study characteristics.**

| First author | Year | Study design | N (M/F) | Age (years) | Objective | Outcomes measured | Sample size estimate |
|---|---|---|---|---|---|---|---|
| | | | | | **Intervention** | | |
| Dahl J [7] | 1993 | R, PD, C/I-2S | 18/0¤ | Md (Rng): 27 (22–39) | Effect of pre- and post-burn lidocaine on hyperalgesia | WDT$_{in/out}$, HPT$_{in/out}$, P-SHA, B-SHA, Flare | NR |
| Møiniche S [22] | 1993 | U, R, OC, 2-A, 1-S, BB | **B₁:** 8/0 **B₂:** 8/0 | **B₁/B₂:** M̄ (Rng): 33 (26–44) | Effect of EMLA (**B₁**) and bupivacaine (**B₂**) on inflammation | **B₁/B₂:** Flare, Blistering | PPA |
| Møiniche S [25] | 1993 | DB, R, PC, 1-S, BB | 12/0 | M̄ (SE): 33 (3) | Effect of local morphine on hyperalgesia | HPT, PPT | NR |
| Møiniche S [24] | 1993 | DB, R, PC, 1-S, BB | 11/1 | M̄ (Rng): 33 (21–46) | Effect of topical piroxicam gel on hyperalgesia and inflammation | HPT, HPTo, P-SHA, MPT, EI, Blistering | NR |
| Brennum J [5] | 1994 | U, R, OC, 2-WX, BB¶¶ | 5/5 | M̄ (Rng): 26 (22–31) | Effect of pre- vs. post-CBI epidural morphine on hyperalgesia | B-SHA, P-SHA, HPT, HPTo, WDT, Flare, ST | NR |
| Møiniche S [26] | 1994 | DB, R, PC, 1-S, BB | 10/2 | Md (Rng): 27 (21–45) | Effect of ketorolac gel on hyperalgesia and inflammation | HPT, HPTo, P-SHA, MPT, EI, Blistering | PPA |
| Pedersen JL [35] | 1994 | DB, R, PC, 1-S, BB | 10/2 | Md (Rng): 31 (23–44) | Effect of topical clobetasol propionate on hyperalgesia and inflammation | HPT, HPTo, P-SHA, MPT, EI, Blistering | NR |
| Ilkjær S [13] | 1996 | DB, R, PC, 3-WX | 19/0 | Rng: 20–31 | Effect of ketamine on hyperalgesia | HPT, P-SHA, B-SHA, CBI-pain | NR |
| Lundell JC [17] | 1996 | DB, R, PC, 1-S, BB | 10‡‡ | NR | Effect of ketorolac on hyperalgesia | HPT, H-pain, CBI-pain | NR |
| Pedersen JL [30] | 1996 | DB, R, PC, 1-S, BB | 12/0 | Rng: 22–47 | Effect of EMLA on late hyperalgesia and inflammation | HPT, P-SHA, MPT, EI, Blistering | PMDD |
| Pedersen JL [31] | 1996 | U, R, OC, 1-S, BB | 18/2¤¤ | Rng: 22–46 | Effect of a preemptive nerve block on late hyperalgesia and inflammation | P-SHA, MPT, CBI-pain, EI, Blistering | NR |
| Warncke T [52] | 1996 | **B₁:** DB, R, PC, 1-S, BB **B₂:** DB, R, PC, 2-WX | **B₁:** 8/2 **B₂:** 8/12 | **B₁:** M̄ (Rng): 24 (21–37) **B₂:** M̄ (Rng): 24 (21–40) | Effects of topical (**B₁**) or oral (**B₂**) ibuprofen on hyperalgesia | **B₁/B₂:** HPT, HPTo, P-SHA, | PPA |
| Ilkjær S [12] | 1997 | DB, R, PC, 3-WX | 25/0 | M̄ (Rng): 24 (21–28) | Effect of dextromethorphan on hyperalgesia | HPT, P-SHA, B-SHA, CBI-pain | NR |
| Pedersen JL [36] | 1997 | SB, R, PC, 1-S, BB | 21/3† | R: 22–46 | Effect of a lumbar sympathetic plexus block on hyperalgesia and inflammation | CBI-pain, MPT, HPT, H-pain, P-SHA, B-SHA, EI, ST | PMDD |
| Petersen KL [39] | 1997 | DB, R, PC, 2-WX | 20/0 | Rng: 19–31 | Effect of ibuprofen on hyperalgesia | P-SHA, B-SHA, B-pain$_{out}$, CBI-pain | NR |
| Warncke T [53] | 1997 | DB, R, A/IC†††, 2-WX, BB | 10/0 | Md: 22 | Effect of ketamine on hyperalgesia and TS in the SHA compared to lidocaine and placebo | WDT$_{in/out}$, HPT$_{in/out}$, MPT$_{in/out}$, TDT$_{in/out}$, P-SHA, TS$_{M,out}$, ST | NR |
| Warncke T [54] | 1997 | DB, R, PC, 3-WX | 10/2 | M̄ (Rng): 22 (20–29) | Effects of morphine and ketamine on hyperalgesia and TS in the SHA | B-SHA, CDT, CPT, HPT$_{in/out}$, MPT, TDT, P-SHA, ST, WDT, TS$_{M,out}$ | NR |
| Pedersen JL [32] | 1998 | DB, R, SyC, PC, 3-WX | 12/3 | Rng: 26–48 | Effect of local ketamine on hyperalgesia | WDT, HPT, H-pain, P-SHA, MPT, M-pain$_{in/out}$, B-SHA, CBI-pain | PMDD |
| Hammer NA [10] | 1999 | DB, R, PC, 2-WX | 17/3 | Rng: 22–48 | Effect of riluzole on hyperalgesia | HPT, H-pain$_{short}$, H-pain$_{long}$, P-SHA, B-SHA, MPT, M-pain$_{in/out}$, CBI-pain | NR |
| Mikkelsen S [20] | 1999 | DB, R, PC, 3-WX | 23/0 | NR | Effect of naloxone on the anti-hyperalgesic effects of ketamine | HPT, P-SHA, B-SHA | NR |
| Sjölund KF [49] | 1999 | DB, R, PC, 2-WX | 5/5 | Rng: 19–31 | Effect of adenosine on hyperalgesia and detection thresholds | B-SHA, CDT$_{in/out}$, HPT$_{in/out}$, H-pain$_{in/out}$, M-pain$_{in/out}$, MPT$_{in/out}$, P-SHA, WDT$_{in/out}$ | NR |
| Holthusen H [11] | 2000 | DB, R, PC, 3-WX | 6/0 | Rng: 27–43 | Effect of pre- vs. post-CBI lidocaine on hyperalgesia | WDT, HPT, P-SHA, MPT, TDT, CBI-pain, Flare | NR |
| Lillesø J [16] | 2000 | DB, R, PC, SyC, 3-WX | 16/2 | Rng: 22–48 | Effect of morphine on hyperalgesia | WDT, HPT, H-pain, CDT, P-SHA, B-SHA, MPT, M-pain$_{in/out}$, CBI-pain | NR |

*(Continued)*

**Table 2.** (Continued)

| First author | Year | Study design | N (M/F) | Age (years) | Objective | Outcomes measured | Sample size estimate |
|---|---|---|---|---|---|---|---|
| Mikkelsen S [21] | 2000 | DB, R, PC, 3-WX | 24/0 | NR | Effect of oral morphine on hyperalgesia | HPT, P-SHA, B-SHA, CBI-pain | NR |
| Warncke T [55] | 2000 | DB, R, PC, 3-WX | 11/1 | $\bar{M}$ (Rng): 24 (21–29) | Effects of morphine and ketamine on hyperalgesia and detection thresholds | B-SHA, HPT$_{in/out}$, MPT$_{in/out}$, TDT, P-SHA, ST$_{in/out}$, WDT, TS$_{M,out}$ | NR |
| Brennum J [6] | 2001 | DB, R, PC, 3-WX | 24/0 | $\bar{M}$ (Rng): 24 (20–31) | Effect of naloxone on hyperalgesia | B-SHA, P-SHA, HPT, H-pain$_{short/long}$, CBI-pain | PPA |
| Werner MU [60] | 2001 | DB, R, PC, 2-WX | 22/0 | Rng: 20–29 | Effects of gabapentin on hyperalgesia | CBI-pain, CDT, HPT, MPT, M-pain$_{in/out}$, WDT, P-SHA, B-SHA | ASE/PMDD |
| Werner MU [58] | 2002 | DB, R, PC, 2-WX | 22/0 | IQR: 25–26 | Effect of systemic dexamethasone on hyperalgesia and thermal detection thresholds | CBI-pain, CDT, H-pain, HPT, M-pain, MPT, P-SHA, B-SHA, EI, WDT | ASE |
| Werner MU [59] | 2002 | SB, R, ShC, 1-S, BB | 24/0 | NR | Effects of local cooling on hyperalgesia and inflammation | CBI-pain, CDT, HPT, MPT, M-pain$_{in/out}$, P-SHA, EI, ST$_{in/out}$, WDT | ASE |
| Schulte H [47] | 2004 | DB, R, PC, 3-WX | 6/5 | $\bar{M}$ (Rng): 36 (22–50) | Synergistic effects of morphine and ketamine | MPT$_{in/out}$, P-SHA, TS$_{M,in/out}$, TDT$_{in/out}$ | ASE |
| Schulte H [48] | 2005 | DB, R, A/IC§§, 3-WX | 8/8 | $\bar{M}$ (Rng): 29 (25–42) | Effect of morphine and alfentanil on hyperalgesia | MPT$_{in/out}$, P-SHA, TS$_{M,in/out}$ | NR |
| Robertson L [63] | 2007 | DB, R, PC, 1-S, BB | 9/15 | Md (Rng): 25.5 (17–39) | Effect of local fentanyl and naloxone on hyperalgesia | H-pain, HPT, M-pain | NR |
| Stubhaug A [51] | 2007 | DB, R, PC, 3-WX | 12/0 | $\bar{M}$ (SD): 23 (2) | Effect of methylprednisolone and ketorolac on hyperalgesia | CBI-pain, HPT, P-SHA | ASE |
| Ravn P [43] | 2013 | DB, R, PC, 5-WX | 14/13†† | $\bar{M}$ (SD): 25 (4) | Differences in anti-hyperalgesic effects of morphine and buprenorphine; relation between opioid effect and pain sensitivity | WDT, HPT, CDT, P-SHA, MPT, CBI-pain | ASE |
| Andersen LPH [1] | 2015 | DB, R, PC, 3-WX | 29/0 | $\bar{M}$ (CI): 26 (25–28) | Effect of melatonin on hyperalgesia and inflammation | CBI-pain, MPT$_{in/out}$, P-SHA, PPT, PPTo, HPT, WDT, EI, DT | ASE/PPA‡ |
| Rasmussen VM [41] | 2015 | U, R, OC, 2-WX | 17/0 | $\bar{M}$ (CI): 28 (25–30) | Effect of hyperbaric oxygen on hyperalgesia | WDT, HPT, CDT, P-SHA, MPT, EI, CBI-pain | ASE |
| Wahl AM [64] | 2019 | SB, R, OC, 2-WX | 19/0 | Md (Rng): 27.5 (21–56) | Effects of hyperbaric oxygen on hyperalgesia post-CBI | WDT, HPT, CDT, P-SHA, MPT, EI, CBI-pain, DT | ASE |
| **Non-intervention** | | | | | | | |
| Methodological | | | | | | | |
| Møiniche S [23] | 1993 | EXP, 2-S, TR | 8/0 | Md (Rng): 28 (20–46) | Examine time course of primary and secondary hyperalgesia | HPT, HPTo, P-SHA, B-SHA, MPT, EI | NR |
| Pedersen JL [33] | 1998 | EXP, 3-S, TR | 11/1 | Rng: 24–47 | To determine the within-day and between-day reproducibility of the CBI-model | WDT, HPT, H-pain, CDT, P-SHA, B-SHA, MPT, CBI-pain, EI | ASE |
| Yucel A [61] | 2001 | EXP | 7/3¤¤¤ | Rng: 21–34 | Effect of pre- and post-injury heat conditioning on hyperalgesia and inflammation in the CBI-model and the topical/intradermal capsaicin model | B-SHA, SBF$_{in/out}$, P-SHA, ST, Flare, CBI-pain | NR |
| Yucel A [62] | 2004 | EXP, R, 2-S-CAP/CBI | 9/3 | $\bar{M}$ (VA NR): 23.6 (1.8) | Comparison of TS between the CBI-model and the heat/capsaicin model | B-SHA, CBI-pain, EPT-SS$_{out}$, EPT-RS$_{out}$, HPT-SS$_{out}$, HPT-RS$_{out}$, MI-pain$_{in}/_{out}$, M-pain$_{in/out}$, P-SHA, TS$_{M,in/out}$, TS$_{I,in/out}$, TS$_{E,out}$, TS$_{H,out}$, Flare | NR |
| Naert ALG [27] | 2007 | EXP, TR¶¶¶ | 31/27 | $\bar{M}$ (SEM): 26 (0.9) | Characterization and validation of the CBI-model as a tonic heat pain model | HPT, HPTo, H-pain, CBI-pain | NR |
| Bishop T [4] | 2009 | EXP, 3-B | 8/4* | $\bar{M}$ (SEM): 26.6 (2.4) | To compare sensory changes of the UVB model to the CBI-model and topical capsaicin model | HPT, P-SHA, B-SHA, MPT, Flare, SBF | NR |

(*Continued*)

**Table 2.** (Continued)

| First author | Year | Study design | N (M/F) | Age (years) | Objective | Outcomes measured | Sample size estimate |
|---|---|---|---|---|---|---|---|
| Ringsted TK [44] | 2015 | EXP, 2-S, TR | 11/12 | $\bar{M}$ (CI): 23.8 (23.2–24.3) | To compare the SHA between 4 stimulators; to examine inter-day and inter-observer differences of the SHA | P-SHA, CBI-pain | ASE |
| | | | | | Physiological | | |
| Pedersen JL [29] | 1998 | EXP, 1-S | 11/1 | Rng: 23–37 | Comparison of TS in normal and hyperalgesic skin | P-SHA, MPT, CBI-pain, EPT-SS$_{in/out}$, EPR$_{in/out}$, TS$_{E,in}$, TS$_{E,out}$ | PMDD |
| Pedersen JL [34] | 1998 | EXP, 1-S | 12/3 | Rng: 26–48 | To examine the evidence for heat hyperalgesia within the mechanical SHA | HPT, H-pain$_{in/out}$, P-SHA, B-SHA, MPT, M-pain$_{in/out}$, CBI-pain | NR |
| Schulte H [46] | 2004 | EXP, 1-S | 9/11 | $\bar{M}$ (R): 37 (19–58) | Examine correlation between SBF and P-SHA | SBF$_{area}$, P-SHA*, ST$_{in/out}$, Flare | NR |
| Norbury TA [28] | 2007 | EXP, 1-S** | 0/196 | Rng: 19–76 | To investigate the heritability of pain sensitivity in a twin study | HPT, P-SHA, B-SHA, CBI-pain, Flare | ASE |
| Drummond PD [8] | 2008 | EXP, PC, OC | 25‡‡ | Rng: 18–58 | To investigate the effect of noradrenaline efflux in the skin on the sensitivity to hyperalgesia | HPT, H-pain, CBI-pain, EPS | NR |
| Robertson L [45] | 2008 | DB, R, PC, 2-WX, BB | 17/5 | Md (Rng): 19 (17–39) | To investigate the effect of repeatedly immersing a CBI in cold water on heat-pain sensitivity | HPT, ST | NR |
| Drummond PD [9] | 2009 | 2-EG-4-CBI, PC | 10/24*** | Rng: 17–32 | To investigate the nociceptive effects of $\alpha$1- and $\alpha$2-adrenoceptor agonists with and without prior $\alpha$-adrenergic blockade | HPT, H-pain | NR |
| Kupers R [15] | 2009 | EXP | 12/9§ | $\bar{M}$ (SD): 32.2 (8.9) | To investigate the correlation between 5-HT$_{2A}$ receptor availability and tonic and phasic heat pain using PET | HPT, HPTo, H-pain, CBI-pain | NR |
| Petersen LJ [40] | 2009 | PD, 3-EG, 1-S | **B$_1$**: 6/0 **B$_2$**: 8/0 **B$_3$**: 8/0 | Rng: 26–46 | To study histamine release in human skin by microdialysis technique after a CBI | No QST | NR |
| Kupers R [14] | 2011 | EXP | 13/8 | $\bar{M}$ (SD): 32.6 (8.8) | To investigate the role of the SERT on pain during tonic and phasic heat using PET | HPT, HPTo, H-pain, CBI-pain | NR |
| Pereira MP [38] | 2013 | DB, R, PC, 2-WX | 11/11 | $\bar{M}$ (SD)$_M$: 24.5 (2.0) $\bar{M}$ (SD)$_F$: 23.0 (1.2) | To examine if naloxone reinstates hyperalgesia after the resolution of a CBI | WDT, HPT, P-SHA, MPT, CBI-pain | ASE |
| Asghar MS and Pereira MP [2] | 2015 | EXP | HS: 6/14 LS: 12/8§§ | $\bar{M}$ (SD)$_{HS}$: 24.3 (2.3) $\bar{M}$ (SD)$_{LS}$: 24.2 (2.6) | To compare structural and functional characteristics of brain activity and anatomy in HS and LS using fMRI | P-SHA, M-pain$_{in/out}$, CBI-pain | ASE |
| Pereira MP [37] | 2015 | DB, R, PC, 2-WX | 12/0 | $\bar{M}$ (CI): 23.8 (22.8–24.9) | To examine if naloxone reinstates hyperalgesia after the resolution of a CBI | WDT, HPT, P-SHA, B-SHA, MPT, CBI-pain | ASE/PSE |
| Slimani H [50] | 2018 | SB, R, ShC, PD | 20/0 | $\bar{M}$ (VA NR)$_{EG}$: 23.8 (1.5) $\bar{M}$ (VA NR)$_{CTRL}$: 24.5 (4.5) | Assess changes in reaction time and sensory detection thresholds in A$\delta$- and C-fibers after a CBI | CBI-pain, HPT$_{AF,in/out}$, HPT$_{CF,in/out}$, H-pain$_{AF,in/out}$, H-pain$_{CF,in/out}$, P-SHA, RTFD$_{AF,in/out}$, RTFD$_{CF,in/out}$ | NR |
| | | | | | Predictive | | |
| C$_1$: Werner MU [57] C$_2$: Werner MU [56]¶ | **C$_1$**: 2003 **C$_2$**: 2004 | **C$_1$**: EXP, PD, 2-S **C$_2$**: EXP | **C$_1$**: Pt 14/4, CTRL: 13/1 **C$_2$**: 14/6 | **C$_1$**: Md (IQR)$_{Pt}$: 27 (25–32); Md (IQR)$_{CTRL}$: 31 (27–40) **C$_2$**: Md (IQR): 28 (24–33) | **C$_1$**: To compare the hyperalgesic response before and after arthroscopic knee surgery **C$_2$**: Effect of pre-operative hyperalgesic response to predict postoperative pain ratings in knee surgery patients | **C$_1$/C$_2$**: CBI-pain, H-pain, HPT, M-pain, MPT, P-SHA | **C$_1$**: ASE **C$_2$**: NR |
| Ravn P [42] | 2012 | EXP, 2-S | 50/50 | $\bar{M}$ (SD): 23.7 (3.6) | Evaluation of QST, psychometrics, gender, and anthropometrics ability to predict pain during a CBI | WDT, HPT, CDT, P-SHA, MPT$_{in/out}$, CBI-pain | NR |

(*Continued*)

**Table 2.** (Continued)

| First author | Year | Study design | N (M/F) | Age (years) | Objective | Outcomes measured | Sample size estimate |
|---|---|---|---|---|---|---|---|
| Lunn TH [18] | 2013 | Prospective, consecutive, observational, cohort | 47/50 | M̄ (Rng): 66 (39–89) | To assess if short and long heat stimulation can predict postoperative pain after total knee arthroplasty | CBI-pain, H-pain | ASE |
| Miscellaneous | | | | | | | |
| Matre D [19] | 2006 | PD, 3-S, PAS | PL: 10/9 CTRL: 7/3 | Rng$_{PL}$: 20–44 Rng$_{Con}$: 20–45 | Effect of a placebo condition on hyperalgesia | P-SHA, B-SHA, MPT, M-pain, CBI-pain | NR |
| Aslaksen PM [3] | 2008 | R, 2-S, OC, PAS | 31/32 | M̄ (SD)$_M$: 25.4 (5.4) M̄ (SD)$_F$: 23.1 (4.7) | Effect of a placebo condition on pain during a CBI and negative emotions | CBI-pain | NR |

**A/IC** = active and inactive placebo-controlled; **AF** = A-fibers; **ASE** = a priori sample size estimate; **B-pain** = motor brush-evoked pain ratings; **B-SHA** = brush secondary hyperalgesia area (allodynia); **CBI** = contact burn injury; **CBI-pain** = CBI-induced pain ratings; **BL** = baseline; **C/I-2S** = 2 groups (1 control and 1 intervention) with 2 sessions per group; **CDT** = cool detection threshold; **CF** = C-fibers; **CI** = 95% confidence interval; **DB** = double-blind; **DT** = dermal thickness; **EG** = experimental group; **EI** = erythema index; **EPR** = electrical pain response to a single stimulus; **EPS** = electrically-evoked pain sensations; **EPT** = electrical pain threshold; **CTRL** = control; **EPT-RS** = electrical pain threshold repetitive stimuli; **EPT-SS** = electrical pain threshold single stimulus; **EXP** = exploratory; **F** = female; **Flare** = area of flare; **fMRI** = functional magnetic resonance imaging; **H-pain** = heat-evoked pain rating; **HPT** = heat pain threshold; **HPTo** = heat pain tolerance; **HS** = high-sensitizers; **I** = intervention; **IQR** = interquartile range; **LS** = low-sensitizers; **M̄** = mean; **M** = male; **M-pain** = punctate mechanical-evoked pain rating; **MI-pain** = mechanical impact-evoked pain ratings; **Md** = median; **MPT** = mechanical pain threshold; **NA** = not applicable; **NR** = not reported; **OC** = open control; **P-SHA** = punctate secondary hyperalgesia area; **PAS** = placebo analgesia study; **PC** = placebo-controlled; **PD** = parallel design; **PET** = positron emission tomography; **PL** = placebo; **PMDD** = post hoc minimal detectable differences; **PPA** = post hoc power analysis; **PPT** = pressure pain threshold; **PPTo** = pressure pain tolerance; **PSE** = post hoc sample size estimate; **Pt** = patients; **QST** = quantitative sensory testing; **R** = randomized; **Rng** = range; **RTFD** = reaction time frequency distribution; **SB** = single-blind; **SBF** = skin blood flow; **SE** = standard error; **SERT** = serotonin transporter; **SHA** = secondary hyperalgesia area; **ShC** = sham-controlled; **ST** = skin temperature; **SyC** = systemic-controlled; **TDT** = tactile detection threshold; **TR** = test-retest; **TS$_M$/TS$_E$/TS$_I$/TS$_H$** = temporal summation to punctate mechanical/electrical/mechanical impact/heat stimulation; **U** = unblinded; **VA** = variance; **WDT** = warmth detection threshold; **1-S/2-S/3-S** = one-/two-/three-session study; **2-A** = 2 arm study design; **2-EG-4-CBI** = 2 experimental groups with 4 burn injury sites in each group; **2-S-CAP/CBI** = 2 sessions with intradermal capsaicin in one session and a burn injury in the other; **2-WX/3-WX/4-WX/5-WX** = two-/three-/four-/five-way crossover design; **3-B** = 3 block study; **3-EG** = 3 experimental groups.

¤ n = 8/12 in control/intervention study; 2 subjects excluded from intervention study.

¤¤ 6 subjects excluded from hyperalgesia comparisons.

¤¤¤ only subjects from burn injury group.

† n = 16 after exclusion.

†† subjects selected based on highest and lowest pain scores during a burn injury in another study.

††† active placebo: Lidocaine (SC 30 mg).

‡ post hoc power analysis performed because of incorrect a priori sample size estimate.

‡‡ total subjects, gender not reported.

§ 3 subjects withdrew from analysis.

§§ categorization into high- and low-sensitizers based on SHAs after a burn injury on the screening day.

§§§ active placebo: Midazolam (IV 20 µg/kg).

ꟸ study C$_2$ included 17 of the same patients as the companion paper study C$_1$.

ꟸꟸ burn injuries were not induced at the same time-point.

ꟸꟸꟸ only 12 subjects retested 3 months later.

* only subjects from study block with burn injury.

** variability and reliability were tested in 10 individuals on two separate days.

*** α1-adrenoceptor (n = 19); α2-adrenoceptor (n = 15). A subscript was added to the different tests to indicate whether they were measured in the primary hyperalgesia area (in) or the SHA (out). **A$_{1-4}$**: Numbering applied to this letter indicates different interventions in the same study group of a particular study. **B$_{1-3}$**: Numbering applied to this letter indicates different interventions or experimental setups in different study groups of the same study. **C$_{1-2}$**: Numbering applied to this letter indicates different companion papers investigating parts or all of the same study group.

drugs to examine any opioid-mediated effects [16,20,63]. Multiple drug dosing was evaluated in seven studies [1,6,12,13,21,43,48].

Methodological quality was assessed by the Oxford Quality Scoring System [75]. The median (IQR) score for the 37 intervention studies was 2 (2 to 3). Fifteen studies had a high-quality score ≥ 3 (Table 3) [1,6,11,17,20,21,41,43,48,51,52,54,55,58,64].

*Methodological studies.* The seven studies [4,23,27,33,44,61,62] focused on characterizing the CBI as a model of tonic heat pain [27]; validating reproducibility [33]; evaluating the vascular and sensory effects of heat conditioning [61]; determining time-courses of hyperalgesia [23]; characterizing SHA assessments [44], and comparing the CBI-model to other models of evoked hyperalgesia [4,62].

The total number of per-protocol subjects was 135, with a median of 12 (11 to 18) subjects per study. The gender ratio (males/females) was 1.7 (85/50). Only one study included data on subjects' height, weight, or BMI [44]. Two studies reported a priori sample size estimates [33,44].

*Physiological studies.* The fourteen studies [2,8,9,14,15,28,29,34,37,38,40,45,46,50] focused on comparing SHAs to areas of flare [46], the adrenergic system [8,9], the endogenous opioid system [37,38,45], heritability of pain responses [28], imaging [2,14,15], latent sensitization [37,38], local inflammatory mediators [40], nociceptive fibers [50], secondary hyperalgesia to heat [34], and temporal summation [29].

The total number of per-protocol subjects was 482, with a median of 21.5 (20 to 24) subjects per study. One study did not report the gender of subjects [8], but for the remaining studies, a gender ratio (males/females) of 0.6 (167/290) was obtained. Three studies provided height and weight of subjects [2,37,38], and one also reported BMI [37]. Four studies reported a priori sample size estimates [2,28,37,38], and one of these included a post hoc sample size estimate [37]. One study calculated post hoc minimal detectable differences for all outcomes [29].

*Predictive studies.* Three of the four predictive studies focused on patients undergoing knee surgery [18,56,57]: Two of these studies investigated the predictive potential of the CBI model [18,56] while the third investigated how surgery modulates hyperalgesic responses [57]. The fourth study evaluated the predictability of pain during a CBI, based on QST assessments [42].

The three knee surgery studies involved patients [18,56,57], while the fourth study included healthy subjects [42]. Accounting for duplicate subjects, the total number of per-protocol subjects was 232. The ratio between male and female subjects was 1.1 (124/108). One study provided height, weight, and BMI of subjects [42]. Two of the four studies reported a priori sample size estimates [18,57].

## Study outcomes

**Intervention studies.** This subsection focuses on the efficacy of interventions in attenuating hyperalgesia, pain, and other inflammatory responses following a CBI. An overview of the intervention studies is presented in Table 3. In general, no intervention showed clear efficacy in attenuating CBI-induced hyperalgesia.

NMDA receptor antagonists: Ketamine reduced the SHA in four studies [13,53–55], while two studies found transient to no effects on hyperalgesia [21,32]. The effects of ketamine on SHAs were observed regardless of whether it was administered preemptively [13,53,55] or post-CBI [54]. Furthermore, ketamine reduced the SHA regardless of a preceding naloxone infusion, indicating that the effect of ketamine was not mediated by the endogenous opioid system [20]. Ketamine was claimed to act synergistically with morphine in reducing temporal summation [47]. Dextromethorphan, another NMDA receptor antagonist, marginally reduced the SHA [12].

**Table 3. Intervention study results.**

| First author | Intervention type | Time of intervention | Specific outcomes | Main outcomes | Quality score |
|---|---|---|---|---|---|
| **Pharmacodynamic** | | | | | |
| *Antiarythmic agents* | | | | | |
| Sjölund KF [49] | Adenosine (IV/7.2 mg/kg) | -15′ | ↓P-SHA, →B-SHA, →$\|CDT_{in/out}\|$, →H-pain, →$HPT_{in/out}$, →M-pain, →$MPT_{in/out}$, →$WDT_{in/out}$ | Adenosine reduced SHAs without reducing primary hyperalgesia | 2 |
| *Gabapentinoids* | | | | | |
| Werner MU [60] | Gabapentin (oral/1200 mg) | -3h | ↑MPT, →B-SHA, $\|$→CDT$\|$, →HPT, →M-$pain_{in/out}$, →P-SHA, →WDT | Gabapentin increased only MPT without decreasing M-pain | 2 |
| *Glucocorticoids* | | | | | |
| Pedersen JL [35] | Clobetasol propionate (topical/0.2–0.3 g 0.05%) | -60′ | →HPT, →HPTo, →P-SHA, →MPT, →EI, →Blistering | Topical glucocorticoid had no effect on hyperalgesia or inflammation | 2 |
| Werner MU [58] | Dexamethasone (IV/8 mg) | ~-2h | →B-SHA, $\|$→CDT$\|$, →H-pain, →HPT, →M-pain, →MPT, →P-SHA. →EI, →WDT | Dexamethasone did not reduce pain, sensory thresholds, or erythema | 4 |
| *Glutamate receptor antagonists* | | | | | |
| Hammer NA [10] | Riluzole (oral/100 mg) | -90′ | →HPT, →H-$pain_{short/long}$, →P-SHA, →B-SHA, →MPT, →M-$pain_{in/out}$, →CBI-pain | Riluzole did not reduce primary hyperalgesia or SHAs | 2 |
| *Local anesthetics* | | | | | |
| Dahl J [7] | Lidocaine (SC/5-6 ml 1%) | Pre-CBI: -5′ Post-CBI: 35′ | Pre-CBI injection compared to post-CBI for all ↓P-SHA¤, ↓B-SHA¤, ↓Flare, →$WDT_{in/out}$, →$HPT_{in/out}$ | Pre-CBI lidocaine more effectively reduced SHAs than post-CBI lidocaine but only shortly | 2 |
| Møiniche S [22] | $B_1$: EMLA (topical/2 g 5%) $B_2$: Bupivacaine (SC/8 mL 0.5%) | $B_1$: -90′ $B_2$: -15′ | $B_1/B_2$: →Flare, →blistering | No anti-inflammatory effects of topical EMLA or subcutaneous bupivacaine | 1 |
| Pedersen JL [30] | EMLA (topical/2 g 5%) | 0′ | →HPT, →P-SHA, →MPT, →EI, →blistering | Topical EMLA did not alter hyperalgesia or inflammation | 2 |
| Holthusen H [11] | Lidocaine (IV/317.5 mg) | Pre-CBI: -30′ Post-CBI: 0′ | ↓P-SHA¤¤, →MPT, →TDT, →HPT, →WDT, →Flare, →CBI-pain | Pre-CBI lidocaine reduced SHAs compared to placebo without being superior to post-CBI lidocaine | 3 |
| *Melatonin* | | | | | |
| Andersen LPH [1] | Melatonin ($A_1$: IV 100 mg; $A_2$: IV/10 mg) | -60′ | $A_1/A_2$: →WDT, →HPT, →P-SHA, →$MPT_{in}$, →$MPT_{out}$, →PPT, →PPTo, →CBI-pain, →EI, →DT | Melatonin did not reduce hyperalgesia or inflammation | 5 |
| *Nerve blocks* | | | | | |
| Pedersen JL [31] | Sapheneous nerve block (lidocaine 9 ml 1%) | CBI induced upon complete block | ↓P-SHA, ↑MPT, →EI, →Blistering | A pre-CBI nerve block reduced primary hyperalgesia and SHAs after return of cold sensation | 1 |
| Pedersen JL [36] | Sympathetic lumbar nerve block (bupivacaine 0.5% 10 ml) | ~-45′ | ↑ST, →CBI-pain, →MPT, →HPT, →H-pain, →P-SHA, →B-SHA, →EI | A pre-CBI sympathetic nerve block did not alter pain or hyperalgesia | 2 |
| *NMDA receptor antagonists* | | | | | |
| Ilkjær S [13] | Ketamine ($A_1$: IV/0.49 mg/kg in 150′; $A_2$: IV/0.98 mg/kg in 150′) | -20′ | $A_1$: ↓B-SHA¤¤¤, →CBI-pain, →P-SHA, →HPT $A_2$: ↓CBI-pain, ↓P-SHA, ↓B-SHA, ↑HPT | Ketamine reduced primary hyperalgesia and SHAs more pronounced with high doses | 2 |
| Ilkjær S [12] | Dextromethorphan ($A_1$: oral/60 mg; $A_2$: oral/120 mg) | -120′ | $A_1$: →P-SHA, →CBI-pain, →B-SHA, →HPT $A_2$: ↓P-SHA, →CBI-pain, →B-SHA, →HPT | High dose dextromethorphan slightly reduced SHA without reducing primary hyperalgesia | 2 |
| Warncke T [53] | Ketamine (SC/4.98 mg) | -20′ | Only compared to inactive placebo ↑$HPT_{in}$, ↑$WDT_{in}$††, ↑$MPT_{in}$, ↓P-SHA, ↓$TS_{M,out}$, →$TDT_{in}$ | Local ketamine showed long-lasting inhibition of the development of SHA | 2 |
| Pedersen JL [32] | Ketamine (SC/7.5 mg) | -40′ and -7′ | ↑HPT‡, ↑MPT‡, ↓M-$pain_{in}$†††, ↓CBI-pain, →M-$pain_{out}$, →P-SHA, →B-SHA, →WDT, →H-pain | Subcutaneous ketamine had brief local analgesic effects | 2 |

(*Continued*)

**Table 3.** (Continued)

| First author | Intervention type | Time of intervention | Specific outcomes | Main outcomes | Quality score |
|---|---|---|---|---|---|
| Mikkelsen S [20] | $A_1$: Naloxone (IV/0.9 mg in 30 min) → Ketamine (IV/0.375 mg/kg in 30 min) $A_2$: Saline → Ketamine (IV/0.375 mg/kg in 30 min) | 75´ | $A_1/A_2$: ↓P-SHA, ↓B-SHA, →HPT | Ketamine reduced SHAs regardless of whether naloxone or placebo was infused prior to ketamine | 4 |
| Mikkelsen S [21] | Ketamine ($A_1$: oral/0.5 mg/kg; $A_2$: oral/1.0 mg/kg) | -20´ | $A_1/A_2$: →HPT, →P-SHA, →B-SHA, →CBI-pain | Oral ketamine did not reduce primary hyperalgesia or SHAs | 5 |
| NSAIDs | | | | | |
| Møiniche S [24] | Piroxicam (topical/1 g 0.5%) | -90´ | →HPT, →HPTo, →P-SHA, →MPT, →EI, →Blistering | Piroxicam gel had no effect on hyperalgesia or inflammation | 2 |
| Møiniche S [26] | Ketorolac (topical/0.75 g 1%) | -90´ | →HPT, →HPTo, →P-SHA, →MPT, →EI, →Blistering | Topical ketorolac did not reduce hyperalgesia | 2 |
| Lundell JC [17] | Ketorolac (intradermally/0.3 mg) | ~-50´ | ↓H-pain‡‡, →HPT, →CBI-pain | Ketorolac decreased the heat pain response in the range of 46-49°C | 4 |
| Warncke [52] | Ibuprofen ($A_1$: topical/3 g 5%; $A_2$: oral/600 mg) | $A_1$: 30´ $A_2$: -70´ | $A_1/A_2$: →HPT, →HPTo, →P-SHA | No anti-hyperalgesic effect of topical or oral ibuprofen | 3 |
| Petersen KL [39] | Ibuprofen (oral/600 mg) | -60´ | ↓B-pain$_{out}$, →CBI-pain, →P-SHA, →B-SHA | Oral ibuprofen had weak effect on allodynia | 2 |
| Opioids | | | | | |
| Møiniche S [25] | Morphine (SC/2 mg) | 30´ | ↑HPT, ↑PPT‡‡ | Local morphine reduced primary hyperalgesia, mainly to heat | 2 |
| Brennum J [5] | Morphine (epidural/4 mg) | 30´§ | Compared to control for both pre- and post-CBI assessments ↑WDT, ↓P-SHA, ↓B-SHA, →HPTo | Pre- and post-CBI morphine attenuated SHAs comparatively | 1 |
| Lillesø J [16] | Morphine (SC/2 mg) | -60´ | ↑P-SHA, ↑B-SHA, →CBI-pain, →MPT, →M-pain$_{in/out}$, →WDT, |→CDT|, →HPT, →H-pain | Local injection of morphine may have contributed to hyperalgesia | 2 |
| Schulte H [48] | Morphine ($A_1$: IV/0.14 mg/kg; $A_2$: IV/0.28 mg/kg) $A_3$: Alfentanil (IV/73 µg/kg) | 70´ | $A_1/A_2$: →P-SHA, →MPT$_{in/out}$, →TS $A_3$: ↓P-SHA¶, ↑MPT$_{in/out}$¶, →TS | Alfentanil had anti-hyperalgesic effects, but no statistically significant dose-dependent effects of morphine were demonstrated | 3 |
| Robertson L [63] | $A_1$: Fentanyl (SC/10 µg) $A_2$: + pretreatment with naloxone (SC/80 µg) | 45´ | $A_1$: ↑HPT, →M-pain§§§, →H-pain $A_2$: →HPT, →M-pain, →H-pain | Naloxone blocks the anti-hyperalgesic effects of fentanyl | 2 |
| Ravn P [43] | Buprenorphine ($A_1$: IV/0.3 mg/210 min; $A_2$: IV/0.6 mg/210 min) Morphine ($A_3$: IV/10 mg/210 min; $A_4$: IV/20 mg/210 min) | -140´ | $A_1$: ↑|CDT|¶¶, ↓CBI-pain, →P-SHA, →WDT, →HPT $A_2$: ↑|CDT|¶¶, ↑HPT¶¶¶, ↓CBI-pain¶¶, →P-SHA, →WDT $A_3$: ↓|CDT|, →P-SHA, →WDT, →HPT, →CBI-pain $A_4$: ↑|CDT|, ↓CBI-pain, →P-SHA, →WDT, →HPT | No clear differences between morphine and buprenorphine in anti-hyperalgesia/analgesia profiles; no difference between high and low pain-sensitive subjects | 5 |
| Opioid antagonists | | | | | |
| Brennum J [6] | Naloxone ($A_1$: IV/0.4 mg; $A_2$: IV/10 mg) | 170´ | $A_1/A_2$: →HPT, →H-pain$_{short/long}$, →P-SHA, →B-SHA | No dose of naloxone had effect on primary hyperalgsia or SHAs | 3 |
| Multimodal | | | | | |
| Warncke T [54] | $A_1$: Morphine (IV/0.15 mg/kg) $A_2$: Ketamine (IV/0.15mg/kg) | 50´ | $A_1$: →WDT, →HPT$_{in/out}$, |→CDT|, →CPT, →TDT, →MPT$_{in/out}$, →B-SHA, →P-SHA, →ST, →TS$_{M,out}$ $A_2$: ↓B-SHA, ↓P-SHA, ↓TS$_{M,out}$*, →WDT, →HPT, |→CDT|, →CPT, →TDT, →MPT, →ST | Ketamine, and not morphine, reduced SHAs and the occurrence of TS | 3 |
| Warncke T [55] | $A_1$: Morphine (IV/0.205 mg/kg in 80 min) $A_2$: Ketamine (IV/0.39 mg/kg in 80 min) | -30´ | $A_1$: ↓TS$_{M,out}$, ↓P-SHA, ↓B-SHA, →MPT$_{in}$**, →HPT$_{in/out}$, →WDT, →ST$_{in/out}$, →TDT, →MPT$_{out}$ $A_2$: ↑MPT$_{in}$***, ↑MPT$_{out}$ ¤¤¤¤, ↓TS$_{M,out}$, ↓P-SHA, ↓B-SHA, →HPT$_{in/out}$, →WDT, →ST$_{in/out}$, →TDT | Pre-treatment with morphine or ketamine reduced SHAs and TS | 4 |

(*Continued*)

**Table 3.** (Continued)

| First author | Intervention type | Time of intervention | Specific outcomes | Main outcomes | Quality score |
|---|---|---|---|---|---|
| Schulte H [47] | $A_1$: Morphine (IV/0.1 mg/kg)<br>$A_2$: Ketamine (IV/0.405 mg/kg)<br>$A_3$: Both drugs | 30′ | $A_1$: →P-SHA, →MPT, →$TS_{M,in/out}$<br>$A_2$: ↓P-SHA††††, ↑MPT, →$TS_{M,in/out}$<br>$A_3$: ↑MPT, ↓$TS_{M,in}$†, ↓$TS_{M,out}$, →P-SHA | Ketamine and morphine reduced TS when co-administered, without affecting TS independently; ketamine transiently reduced SHAs | 2 |
| Stubhaug A [51] | $A_1$: Methylprednisolone (IV/125 mg)<br>$A_2$: Ketorolac (IV/60 mg) | 45′ | $A_1$/$A_2$: ↓P-SHA | Both methylprednisolone and ketorolac decreased SHAs | 5 |
| **Non-pharmacodynamic** | | | | | |
| Hyperbaric oxygen | | | | | |
| Rasmussen VM [41] | Hyperbaric oxygen (100% O2, 2.4 ATM for 90 min; ambient pressure as control) | 0′ | ↓P-SHA, →MPT, →CBI-pain, →EI, →WDT, →HPT, \|→CDT\| | Hyperbaric oxygen reduced SHA, with effect derived from the sequence starting with control | 3 |
| Wahl AM [64] | Hyperbaric oxygen (100% $O_2$, 2.4 ATM for 90 min; ambient pressure as control) | 0′ | ↓P-SHA, →SE, →DT, →CDT, →MPT, →CBI-pain, ↓WDT, ↓HPT | Hyperbaric oxygen reduced SHA, with effect derived from the sequence starting with control | 3 |
| Local cooling | | | | | |
| Werner MU [59] | Local cooling via contact thermode (12.5cm$^2$, 8°C, 30 min, 6.5kPa; dummy thermode as control) | 8′ | ↑\|CDT\|§§, ↓$ST_{out}$§§, →HPT, →M-Pain$_{in/out}$, →MPT, →P-SHA, →EI, →$ST_{in}$, →WDT | Local cooling did not reduce inflammation or hyperalgesia | 2 |

**B-pain** = motor brush-evoked pain ratings; **B-SHA** = brush secondary hyperalgesia area (allodynia); **CBI** = contact burn injury; **CBI-pain** = CBI-induced pain ratings; **CDT** = cool detection threshold; **DT** = dermal thickness; **EI** = erythema index; **EMLA** = Eutectic mixture of local anesthetics; **EPR** = electrical pain response to a single stimulus; **EPS** = electrically-evoked pain sensations; **EPT** = electrical pain threshold; **EPT-RS** = electrical pain threshold repetitive stimuli; **EPT-SS** = electrical pain threshold single stimulus; **Flare** = area of flare; **H-pain** = heat-evoked pain rating; **HPT** = heat pain threshold; **HPTo** = heat pain tolerance; **M-pain** = punctate mechanical-evoked pain rating; **MI-pain** = mechanical impact-evoked pain ratings; **MPT** = mechanical pain threshold; **P-SHA** = punctate secondary hyperalgesia area; **PPT** = pressure pain threshold; **PPTo** = pressure pain tolerance; **SBF** = skin blood flow; **SHA** = secondary hyperalgesia area; **ST** = skin temperature; **TDT** = tactile detection threshold; $TS_M$/$TS_E$/$TS_I$/$TS_H$ = temporal summation to punctate mechanical/electrical/mechanical impact/heat stimulation; **WDT** = warmth detection threshold.

¤ reduced by pre- compared to post-CBI only at 40′ and 70′ post-CBI, and not 100–190′ post-CBI and not by ANOVA 40–190′ post-CBI.

¤¤ only reduced by pre-CBI lidocaine compared to control.

¤¤¤ only short-term effect during infusion, significant at 110′ post-CBI.

¤¤¤¤ significant only during infusion.

† only significant effect at 15′ post-dosing (45′ post-CBI).

†† only significant at 30′ post-CBI.

††† only significant compared to systemic ketamine and only at 0′post-CBI.

†††† only significant effect at 45′ post-dosing (75′ post-CBI), not 75′ post-dosing.

‡ only significant at 0′post-CBI.

‡‡ no difference compared overall temperatures (P = 0.08), however, in the range of 46-49°C, ketorolac decreased pain response (P < 0.05; ANOVA); no benefit of treatment at high temperatures (e.g., 50°C and 51°C).

‡‡‡ only significant effect when analyzing relative changes non-parametric 2-way ANOVA; only significant at 30′ post-CBI when not analyzing relative changes.

§ only one injection, but this was at -150′ compared to second burn

§§ only significant effect at 10′ post-cooling.

§§§ decreased relative to naloxone treated site, but not pre- vs. post-drug.

ƒ only significant at 155′ post-CBI during the infusion, and not at 275′ post-CBI after the infusion.

ƒƒ significant compared to placebo and low-dose morphine.

ƒƒƒ significant compared to placebo and both morphine doses.

* only significant at 65′ post-CBI, 15′ post-infusion.

** only short-term effect at 0′; not significant overall by ANOVA P = 0.18.

*** only significant until 30′ post-CBI, not at 110′ and 150′ post-CBI. Arrows indicate significant increases (↑) or decreases (↓), or no difference (→) after intervention compared to placebo if not otherwise stated. The **Time of intervention** column provides the timing of intervention administration in relation to the burn injury induction (time 0); negative values thus refer to a pre-CBI administration, and positive values refer to administration post-CBI (including 0′ which is immediately post-CBI).

Opioids: A preemptive infusion [55], but not a post-CBI infusion [47,48,54], of morphine, attenuated SHAs, and temporal summation. Epidural morphine injection reduced the SHA regardless of administration timing [5]. No difference was found between systemic buprenorphine or morphine administration on hyperalgesia, nor between the effect on high- and low pain-sensitizer subjects [43].

Glucocorticoids and NSAIDs: One comparative study found an effect of methylprednisolone and ketorolac, respectively, on reducing the SHA, with no statistical difference between the two drugs [51]. No other effects of glucocorticoids on hyperalgesia and inflammation were reported [35,58]. The effects of NSAIDs on heat pain perception were weak and observed exclusively within the range of 46˚C to 49˚C, but interestingly not at higher temperatures (50˚C to 51˚C) [17]. Pain intensity evoked by motor brush stimulation in the SHA was lower following ibuprofen compared to placebo, while the magnitude of SHA was unaltered [39]. No effect of topical NSAID on hyperalgesia and inflammatory variables has been observed [24,26,52].

Antiarrhythmics: Adenosine infusion reduced the SHA without reducing primary hyperalgesia to mechanical or thermal stimuli [49].

Nerve blocks: A saphenous nerve block was found to reduce primary hyperalgesia and SHAs [31], while a sympathetic nerve block did not alter pain or hyperalgesia [36]. These studies were not double-blinded.

Local anesthetics: Two studies evaluated the effect of administering intravenous [11] and subcutaneous [7] lidocaine pre- vs. post-CBI and found no clear superiority of one over the other [7,11]. EMLA-cream [22,30] and bupivacaine [22] had no clear effect on thermal hyperalgesia [30] nor any inflammatory variables [22,30].

Miscellaneous: Gabapentin increased mechanical pain thresholds but did not affect mechanical pain intensity nor other sensory variables [60]. Two studies evaluated the effects of hyperbaric oxygen treatment on secondary hyperalgesia immediately post-CBI using almost identical experimental designs [41,64]. Both studies found that hyperbaric oxygen treatment reduced the SHA when compared to ambient pressure [41,64]. Interestingly, the sequences starting with hyperbaric oxygen, conditioned the response in the subsequent ambient pressure test day (serving as a control), generating a similar reduction in SHAs. Riluzole (glutamate receptor antagonist) [10], melatonin [1], μ-opioid receptor antagonists [6], and local cooling of the CBI [59] had no clear effect on mechanical or thermal hyperalgesia [1,6,10,59,60], nor on other inflammatory variables [1,59].

*Methodological studies.* Repeatability: Within-day comparisons using the CBI-model (420 s at 47˚C, 12.5 cm$^2$) revealed that a 20% difference could be detected for all variables in a crossover design with twelve subjects ($\alpha = 0.05$, $\beta = 0.20$), except for cool detection threshold and heat pain response at 43˚C (see S1 Table for statistical definitions) [33]. On each day, the CBI produced robust hyperalgesia throughout the 6 h study period. Between-day comparisons revealed that the intra-individual, between-days coefficients of variation ranged from 9 to 36% for all sensory assessments along with erythema, except for the heat pain response to 43˚C, and the brush-evoked allodynia, both showing a low reproducibility. These two variables were recommended to omit in between-day comparison studies [33]. A study using a comparable CBI-model (420 s at 47˚C, 9 cm$^2$) observed a correlation between test-retest of pain intensity scores following CBIs ($r^2 = 0.44$; $P < 0.05$) [27]. A study categorized as an intervention study, tested reliability (intraclass correlation coefficient [ICC]), and variability of areas of flare, secondary hyperalgesia, and allodynia using a relatively milder CBI-model (330 s at 45˚C, 0.3 cm$^2$) [28]. The authors concluded that the model was reliable in an investigator independent manner, although primary hyperalgesia outcomes were not addressed. The testing was performed 15 min post-CBI, consequently only addressing the reproducibility of the hyperalgesic response at a limited time span after the CBI.

Validation: The induction of a CBI produced long-lasting hyperalgesia, with primary hyperalgesia lasting between 24–48 h and secondary hyperalgesia remaining until 24 h post-CBI (300 s at 49˚C, 3.75 cm$^2$) [23]. One test-retest study evaluated the performance of different punctate stimulators for the assessment of SHAs [44]. The delineation of SHAs with nylon filaments and weighted-pin instruments were highly correlated with the application pressure. The weighted-pin instrument applying a high pressure (10,424 kPa) showed the greatest inter-observer reliability following a CBI [44].

A comparison of the CBI-model to other cutaneous heat pain models can be found in S3 File.

*Physiological studies*. A functional magnetic resonance imaging (fMRI) study found that high-sensitizers, i.e., subjects with large SHAs, had less activation of the default mode network (precuneus, posterior cingulate cortex) during noxious punctate mechanical stimulation post-CBI compared to low-sensitizers, i.e., subjects with smaller SHAs. Further, an inverse relationship between the magnitude of SHA and the volume of the caudate nucleus was found [2]. A twin-study including 51 monozygotic and 47 dizygotic twin pairs found that the SHA, area of brush-evoked allodynia, HPT, and pain during the CBI all had statistically significant heritable components. However, neither shared genetics nor environmental factors could explain the extent of CBI-induced heat hypersensitivity (ΔHPT [pre- vs. post-CBI]) [28].

*Predictive studies*. In patients undergoing elective arthroscopic anterior cruciate ligament repair, pain during a preoperative CBI significantly correlated with self-reported dynamic pain on postoperative days 0 to 2 (r = 0.65; P < 0.01) and days 3 to 10 (r = 0.57; P < 0.01) and resting pain ratings on postoperative days 0 to 2 (r = 0.60; P < 0.01) and days 3 to 10 (r = 0.59; P < 0.01) [56]. Other QST-variables such as thermal thresholds, mechanical thresholds, and SHA displayed a relatively weaker predictive value. A later high-powered study investigated the ability of a phasic (5 s) vis-á-vis a tonic (420 s) heat stimulus to predict postoperative pain, using identical stimulus areas (12.5 cm$^2$) and temperatures (47˚C) after total knee arthroplasty [18]. The authors found a significant, but very weak, correlation between postoperative VAS during walking on days 1 to 7, and, the phasic (r = 0.25; P = 0.02) and tonic (r = 0.27; P = 0.01) heating paradigms, respectively, deeming these variables clinically irrelevant predictors. Further, arthroscopic knee surgery did not alter postoperative hyperalgesic responses to CBIs when compared to the baseline preoperative responses [57].

## Discussion

### Short summary

The present review, including 64 studies, indicates that the contact burn injury model is a reference model in human experimental pain research. However, our data demonstrate first, a surprising lack of methodological standardization, despite the limited number of research groups employing the CBI-model in pain research. Second, although the model consistently provides long-lasting primary and secondary hyperalgesia, the reproducibility of the sensitizing responses of the model has only been validated in three studies [28,33,90]. Third, the evidence for using the tonic CBI-stimulus to predict acute post-surgical pain is weak [18,56]. Fourth, the pharmacological intervention studies with anti-inflammatory drugs, e.g., NSAIDs and glucocorticoids, have mostly been negative, which is perplexing, since the CBI is considered an inflammatory model. Fifth, while anti-hyperalgesic effects have been demonstrated for ketamine and to some extent also for adenosine, local anesthetics, and opioids, administered following the CBI, no single pharmacological intervention has been showing a consistent efficacy in reversing CBI-induced hyperalgesia.

## Main contact burn injury model variables

**Paradigms and thermode handling.** The included studies reveal substantial variations in the applied heating paradigms (Table 2). Although the relationship between exposure time and temperature of a burn injury is established (Fig 2) [72], the time-course of hyperalgesia has only been investigated in a few studies, and thus different heating paradigms have not been compared. The heating paradigm of 420 s at 47˚C is, however, the most commonly applied [1,2,5,6,10,12–16,18,20,21,27,29,32–34,36–39,41–44,46,48–50,53,55–62]. Reproducibility has been investigated for this heating paradigm [33], hyperalgesia is long-lasting [33] and adverse events are minor and infrequent (reported in 10/39 studies) [29,32–34,36,39,42,46,49,60].

None of the included studies have investigated the effects of changing the active thermode areas. The SHA has been observed to increase proportionally with the CBI-area (preliminary observations by the authors). Interestingly, a study attempting to replicate the heat/capsaicin model with a smaller thermode than previously reported (9 cm² vs. 12.5–15.7 cm²) [70,73] found that the smaller thermode was not able to produce stable SHAs, and, further, could not show an effect of gabapentin, as previously reported [91]. Similarly, another study could not find an effect of pre- and post-conditioning on heat sensory outcomes in the heat/capsaicin model [61]. These authors also used a smaller thermode during rekindling (3.75 cm²) compared to previous studies [70,73]. The findings indicate that changes in the contact thermode area probably may affect the degree of hyperalgesia and the pharmacological sensitivity of the CBI-method.

Contact thermode application pressures were reported in 34/64 of the included studies, utilizing a wide range of pressures. Interestingly, one study investigated the effect on thermal thresholds of varying application pressures (0.32–10 N) using an active thermode area of 9 cm² [92]. The authors concluded that altering the application pressure did neither affect the thermal thresholds nor affect the intra- or inter-subject reproducibility. However, these data only comprised phasic stimuli, not the tonic CBI-stimulus.

**Induction site.** Differences in sensitivity may occur between different skin sites as normal skin on the volar forearm was significantly more sensitive than the medial calf based on thermal detection thresholds [33] and mechanical pain thresholds [93], although conflicting results were found for heat pain thresholds [33,93].

Repeated CBIs on an ipsilateral, homologous skin site have been observed to induce habituation effects across sessions, e.g., decreased SHAs and increased heat pain thresholds in the second session [33,38,42,64]. This problem may be mitigated by using block-randomization and cross-over designs, thereby evenly dividing active drugs and placebo between sessions.

## Validity and reproducibility of contact burn injury variables

**Pathophysiological changes.** As previously reported, Moritz and Henriques described first-degree burn injuries ranging from transient hyperemia to prolonged erythema with the formation of small vesicles [72]. Erythema reflects vascular changes in the burn area, and the degree of redness is measured by skin reflectance spectrophotometry [94]. The duration of erythema has varied substantially across CBI-studies. Other indices of inflammation, i.e., increased dermal blood flow, a rise in skin temperature, and flare, are only short-lasting phenomena post-CBI [46,61,95–97]. Increased dermal thickness as an index of skin edema, measured by high-frequency ultrasound technique, has only been applied in a limited number of studies, but seem to persist several hours post-CBI [1,64]. Interestingly, none of the included studies reported any significant effect on the classic inflammatory variables (edema, erythema, or temperature increase) by the drugs tested.

Assessing the central anti-hyperalgesic effect, by evaluating changes in SHA, is an essential outcome of many analgesic studies. The SHA is delineated via punctate stimulation with a nylon filament or a weighted-pin instrument. The CBI induces rather consistent secondary hyperalgesia, with a duration of 24 h. Interestingly, two studies have indicated that 12/100 (420 s at 47°C, 12.5 cm$^2$) [42], and 10/64 (300 s at 46°C, 12.5 cm$^2$) [19] of the subjects do not develop measurable SHAs. Paradoxically, QST assessments by themselves may induce secondary hyperalgesia [33,98].

Temporal summation, another measure of central sensitization, is generally facilitated in the CBI-model [47,48,53–55,62]. The temporal summation response appears modality-dependent due to the stimulation of different receptor subtypes [29,62]. Temporal summation has primarily been investigated in studies using ketamine and morphine, with the former usually providing a significant mitigating effect. However, other analgesics, e.g., dextromethorphan [99], imipramine [100], venlafaxine [101], and gabapentin [102], have been found to mitigate temporal summation, and could thus also be applied in the CBI-model.

**General issues.** Only a few studies have investigated test-retest reproducibility of the CBI-model variables, and when tested, the applied statistical methods have been inconsistent [27,28,33,44]. When calculating ICCs as a measurement of reliability, the between-subject variance and within-subject variance should preferably be reported, since the former may increase as a result of measurement heterogeneity on a group level, thereby artificially increasing ICCs [103]. However, the included studies did not provide this information [28,44]. An agreement test, presented as a Bland-Altman plot, has been recommended as a valid test-retest statistic mainly because of its detailed, transparent, and honest presentation of data distribution [103]. However, only one of the included studies presented this statistic, and the authors mainly focused on the inter-observer agreement [44]. One study claimed that there was a good test-retest reproducibility of pain scores during a CBI [27], but this was only based on a correlation coefficient between test-retest scores which is a suboptimal measure of reproducibility [103–105]. A review providing a post hoc analysis of test-retest data from a CBI-study found that SHAs were reproducible across multiple sessions. These calculations were based on ICCs, suggesting that subjects could be phenotyped according to the pattern of sensitization [42,90].

When assessing quantitative sensory thresholds, the subject's reaction time, depending on nerve conduction velocity and cognitive-executive abilities, may influence the results. However, only a few of the included studies reported reaction time tests [1,5,43]. Sensory tests applying the 'method of limits' are especially subjected to the influence of reaction time. The effect of analgesic drugs with potential sedative properties may increase response latency, erroneously leading to higher pain thresholds ('pseudoanalgesia') [43]. This is an underreported source of systematic bias.

## The contact burn injury in pharmacodynamic research

None of the evaluated analgesics had a consistent pharmacodynamic profile of attenuating, either primary or secondary hyperalgesia. Further, a substantial number of studies received a low-quality score questioning the methodological quality of the studies (cf. Methodological limitations of the review). Ketamine administration, however, was associated with a reduction in SHA and temporal summation, thus clearly attenuating central sensitization phenomena [53–55]. The effects of ketamine on primary mechanical hyperalgesia were more pronounced than on primary thermal hyperalgesia, which may be explained by potentially different mechanisms for mechanical and thermal hyperalgesia [34].

The administration of ibuprofen was associated with a reduction in motor brush-evoked pain response in the SHA following a CBI [39]. An fMRI study found that ibuprofen only

induced cerebral blood flow changes during ongoing postoperative pain, not evidenced in pain-free subjects [106]. The results from the NSAID studies indicate that the effects are primarily centrally mediated but that relatively large doses are needed to reveal this effect [51]. Likewise, the one study that demonstrated an effect of glucocorticoids on SHAs [51] applied a larger, supra-physiological dose than used in previous studies [58]. The central effect of this large dose of glucocorticoid was rapid, thus likely not solely dependent on translational mechanisms. Interestingly, the studies evaluating anti-inflammatory drugs, i.e., NSAIDs and glucocorticoids, have mostly been negative but have received varying quality scores, raising the question of whether the results are correlated with the quality of the study. However, of the six studies evaluating NSAIDs, three different drugs and four different routes of administration were used along with several different outcome measures, obstructing a valid comparison between the studies. Further, among the three high-quality studies, only one study could be considered yielding a positive outcome [51], thus not changing the general conclusion of this review.

Regarding opioids, experimental pain models that apply deep tonic stimulations, thus primarily activating C-fibers, have been found to be more sensitive to systemic morphine than superficial phasic stimulations [107]. Compellingly, opioids reduce pain scores during a CBI [43], indicating that nociceptive transmission is at least partially C-fiber mediated.

Interestingly, although gabapentin only just significantly increased post-CBI mechanical pain thresholds [60], the number of clinical studies investigating this drug increased markedly in the following decades [108]. This exemplifies the use of the CBI-model as a translational model from experimental to clinical research.

## The contact burn injury in clinical predictive studies

The potential for the CBI, applied as a test stimulus or a conditioning stimulus, in predicting acute postoperative pain, has been a matter of debate [18,56]. These discrepancies are likely due to the investigation of different surgical procedures (arthroscopic knee surgery vs. total knee arthroplasty), different test-sites (calf vs. thigh), and that a majority of patients in the negative study presented with preoperative pain [18]. Further, a type I error may have occurred in the hypothesis-generating study [56] since the negative follow-up [18] study included nearly five times as many patients than the positive study.

## Methodological limitations of the review

The methodological quality of the intervention studies, assessed by the Oxford Quality Scoring System [109], was generally fairly low with a median (IQR) value of 2 (2–3). The Oxford Quality Scoring System was chosen over more sophisticated tools (e.g., Cochrane Collaboration's tool for assessing the risk of bias), first, in awareness of the studies varying bibliographical age, and second, because it displays a high interrater validity [109]. Intervention studies were evaluated regardless of their quality score. The quality score depends on the information available in the reports, and journal requirements for trial reporting have not always been as comprehensive as today [109]. Furthermore, the scoring systems used in systematic reviews and meta-analyses, albeit important, only characterize certain aspects of the research design [110,111]. The quality scorings, therefore, may be difficult to interpret in a review, including studies of different time epochs. Thus, the individual studies may have been performed well regarding the somatosensory testing methodology, the main objective of the present review, nevertheless, receiving a low-quality rating. Valuable methodological information could have been neglected by excluding studies based on quality scores. Attempts were made at performing a quality scoring of the non-interventional studies. Incorporating study quality could provide

clarity in studies presenting with contrasting results. Two potential scoring systems were considered: the GRADE tool [112,113] and the "checklist for the assessment of the methodological quality" [114]. However, due to considerable heterogeneity across the studies regarding methodology, outcome variables, and statistical processing, the authors decided to abandon the attempt to make a relevant quality scoring.

Another essential point is that the included studies demonstrated a high heterogeneity regarding the methodology of the CBI-model and associated outcome assessments. Consequently, a statistical meta-analysis of the included results was not deemed feasible and was not within the scope of this review.

When evaluating the effects of analgesics, and especially when comparing several analgesics, it is important to study the dose-response relationship [115]. However, only 20% of the pharmacodynamic studies used more than one drug dose, precluding examination of dose-response relationships.

## Essential elements in future research

**The heating area as an important variable.** While the exposure time and the contact thermode temperature are well-known critical variables in the thermal energy transfer, the size of the heating area has not been systematically examined. The active thermode area governs the intensity of pain during CBI-induction and the primary hyperalgesia area, but also likely influences the magnitude of the SHA, the duration of secondary hyperalgesia, and sensory perception within the hyperalgesia areas. Hypothetically, these spatial characteristics may affect the pharmacodynamic effects of relevant analgesics.

**The contact burn injury as an inflammatory model.** The CBI is a surrogate, experimental model of inflammatory pain, and therefore demonstrates certain limitations compared to clinical pain. Although the model involves the cardinal signs of inflammation, i.e., edema, erythema, local hyperthermia, and evoked pain, the pharmacological sensitivity to anti-inflammatory analgesics, NSAIDs, and glucocorticoids, is ambiguous. The model lacks the spontaneous pain component usually present in acute clinical inflammatory states. A hypothetical explanation is that the area of the skin injury, corresponding to 0.06% of the body surface area, only activates a limited part of the mesencephalic-subcortical pain network. The limited nociceptive drive likely affects the response to relevant analgesics. Obviously, augmenting the extent of thermal injury by increasing the exposure time and/or the contact thermode area/temperature would lead to a second-degree CBI at the undesirable cost of irreversible skin damages. However, the inflammatory response could instead be augmented synergistically by topical administration of pro-inflammatory agents such as mustard oil [116], sodium lauryl sulfate [117], or the well-investigated capsaicin [73], enhancing the primary and secondary hyperalgesia components.

**Characterization of phenotypes.** As previously mentioned, phenotyping based on the SHA is possible across individuals [90]. These phenotypes are characterized by differences in brain structure and function [2], with 'high-sensitizers' presenting similar brain activation patterns to chronic pain patients [2]. Phenotyping based on a subject's 'sensitization' pattern assessed by QST provides a potential for predicting the pharmacodynamic response to analgesics [118–120].

## Conclusion

First, the present review revealed a large heterogeneity in the applied methodologies across the use of the CBI-model. It is well-known how the exposure time and temperature influence the degree of the cutaneous injury (Fig 2). However, another principal variable governing the heat energy transfer, the active area of the contact thermode, has not been systematically examined.

Second, the pharmacodynamic sensitivity of the CBI-model suggests that the paradigm still needs to be optimized. The model shows robust analgesic efficacy for ketamine, but generally lacks sensitivity for anti-inflammatory drugs, contradicting the epithet that the CBI is an inflammatory pain model. Third, the evidence base for an optimal CBI-model is inadequate. However, standardization of the model is needed, and consequently, this review has provided suggestions for future use of the model (Table 4).

**Table 4. Suggestions to consider when designing a CBI-model trial.**

**CBI-methodology**
 1) Equipment calibration
 2) Heating paradigm: 420 s at 47.0˚C; $\geq 9$ cm$^2$
 3) Induction site: lower arm or leg

**Outcome assessments**
 4) Pain during the CBI
 5) Primary hyperalgesia
 • Mechanical/thermal thresholds
 • Mechanical/thermal suprathreshold stimuli
 • Temporal summation
 6) Secondary hyperalgesia
 • Mechanical delineation of SHAs
 • Temporal summation
 7) Inflammatory variables
 • Erythema
 • Edema

**Pharmacodynamic considerations**
 8) Measurement of subjects' reaction times ('method of limits')
 9) Use suitable dose-response designs

**Statistical considerations**
 10) Consider obtaining test-retest data

Authors' considerations, based on the review, when designing a contact burn injury (CBI)-model trial. **1**) Calibration of the CBI-equipment is mandatory. **2**) The most commonly used heating paradigm (the most commonly used heating area was 12.5 cm$^2$). **3**) The induction sites are the most frequently used. **4**) CBI-induced pain can be used both as conditioning- and test-stimulus. **5**) In the CBI-area assessment of thresholds, pain perception to suprathreshold stimuli or temporal summation can be used. **6**) The secondary hyperalgesia area (SHA) is delineated by mechanical stimuli [44], and perception to temporal summation can be evaluated as central sensitization measures. **7**) CBI-induced inflammatory variables to be considered are erythema and edema [1]. **8**) Analgesic trials, using the 'method of limits' in threshold assessments, may require assessment of the subjects' reaction times in order to avoid 'pseudo-analgesia' [43]. **9**) Dose-response studies require a design with multiple doses [115]. **10**) In order to validate the study methodology, a test-retest set-up should be considered [103].

## Supporting information

**S1 Checklist. PRISMA 2020 checklist.**
(DOCX)

**S1 Table. Statistical definitions.**
(DOCX)

**S1 File. Embase search terms.**
(DOCX)

**S2 File. Results from the miscellaneous studies.**
(DOCX)

**S3 File. The contact burn injury model in comparison to other cutaneous heat pain models.**
(DOCX)

## Author Contributions

**Conceptualization:** Anders Deichmann Springborg, Mads Utke Werner.

**Formal analysis:** Anders Deichmann Springborg, Caitlin Rae Wessel, Mads Utke Werner.

**Investigation:** Anders Deichmann Springborg, Caitlin Rae Wessel.

**Methodology:** Anders Deichmann Springborg, Caitlin Rae Wessel, Mads Utke Werner.

**Project administration:** Anders Deichmann Springborg.

**Supervision:** Mads Utke Werner.

**Validation:** Anders Deichmann Springborg, Caitlin Rae Wessel, Lars Peter Kloster Andersen, Mads Utke Werner.

**Visualization:** Anders Deichmann Springborg, Caitlin Rae Wessel, Mads Utke Werner.

**Writing – original draft:** Anders Deichmann Springborg, Caitlin Rae Wessel, Mads Utke Werner.

**Writing – review & editing:** Anders Deichmann Springborg, Caitlin Rae Wessel, Lars Peter Kloster Andersen, Mads Utke Werner.

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
