## [Decision Letter · Decision Letter 0]

19 Feb 2021

PONE-D-20-38717

Methodology and applicability of the human contact burn injury model: A systematic review

PLOS ONE

Dear Dr. Springborg,

Thank you for submitting your manuscript to PLOS ONE. After careful consideration, we feel that it has merit but does not fully meet PLOS ONE’s publication criteria as it currently stands. Therefore, we invite you to submit a revised version of the manuscript that addresses the points raised during the review process.

We look forward to receiving your revised manuscript.

Kind regards,

John Kramer

Academic Editor

PLOS ONE

Journal Requirements:

Additional Editor Comments:

Thank you for submitting your manuscript to PLOS ONE. The comments from the reviewers are attached. Overall, the manuscript has potential to serve as a valuable resource for researchers interested in the contact burn injury model of inducing primary and secondary hyperalgesia. My primary concern, however, is that the manuscript is burdened by redundancies, which makes it rather overwhelming to read and interpret. I would ask that the authors consider the comments of the reviewers and in turn attempt to streamline the results section accordingly, condensing where possible.

In addition, I would request that the authors confirm they searched for previously published systematic reviews/meta-analyses on the topic of contact burn injuries. This needs to be done to avoid duplicate efforts and multiple reviews written needlessly on the same topic. If there are none, please indicate as such in the manuscript. Finally, please consider using the AMSTAR tool to evaluate your systematic review and ensure it meets the highest methodological standard.

Reviewers' comments:

Reviewer's Responses to Questions

**Comments to the Author**

1. Is the manuscript technically sound, and do the data support the conclusions?

Reviewer #1: Partly

Reviewer #2: Yes

2. Has the statistical analysis been performed appropriately and rigorously? 

Reviewer #1: N/A

Reviewer #2: Yes

3. Have the authors made all data underlying the findings in their manuscript fully available?

Reviewer #1: Yes

Reviewer #2: Yes

4. Is the manuscript presented in an intelligible fashion and written in standard English?

Reviewer #1: Yes

Reviewer #2: Yes

5. Review Comments to the Author

Reviewer #1: This manuscript provides a very thorough overview and summary of the contact burn injury model. The background section of the introduction is well written and succinct, however following sections providing additional background are lengthy and could be condensed and incorporated into the introduction, methods, and discussion. The methods are well written, however, substantially more information is needed on the quality assessment of articles. The results are rather repetitive, and the discussion is lengthy, and could be condensed. My main concern with the manuscript is the lack of removal of low-quality intervention studies. It is unclear to what end the author’s main findings are derived from low quality studies compared to high quality. I would suggest incorporating a more critical assessment of the literature throughout the discussion.

Specific comments attached separately.

Reviewer #2: This systematic review by Springborg and colleagues aimed to synthesize data from studies using human burn injury models. This is an important work and, for the most part, provides a nice overview of many human model parameters as applied in the literature. It is very comprehensive, which introduces some difficulties in reporting clear, concise and synthesized results due to obvious variation between studies.

I only have a few comments:

Page 12, line 125: "QST…….by application of graded chemical, electrical,…." What chemical and electrical stimuli are the authors referring to? To the reviewer`s knowledge, classical QST protocols do not employ such stimuli.

I would like to commend the authors on their consistent use of terminology, as they are certainly well aware of the ongoing controversy regarding 'central sensitization' being used as a broad term in human studies in contrast to the very well-defined neuronal construct in basic animal studies. I am glad to see that the psychophysical terminology was used and distinctions between primary and secondary hyperalgesia were made appropriately. This was especially important for the interventional trials as some drugs will affect the primary input and the 2HA, respectively.

However, with respect to the "timing of intervention" it would be interesting to know if the studies aimed at investigating whether the drugs affected the induction of hyperalgesia or lead to a reversal of an already established state of hyperalgesia? In my personal opinion, only the former would constitute a true "anti-hyperalgesic effect". The latter could just be called "analgesic effect" considering CBI as a human surrogate pain model.

In table 6 the use of "assessments of central mechanisms" is recommended. These tests are merely listed in the outcomes section in the introduction.

Please explain the methods/paradigms in a bit more detail, especially with regard to what mechanisms are potentially investigated with the respective paradigms.

Thank you.

6. PLOS authors have the option to publish the peer review history of their article (what does this mean?). If published, this will include your full peer review and any attached files.

Reviewer #1: No

Reviewer #2: No

---

## [Author Response · Author response to Decision Letter 0]

4 Apr 2021

Below you will find a condensed answer to the requested corrections from the Reviewers. A complete response to the Editor and Reviewers' queries can be found in the 'Response to Reviewers' letter. 

COMMENTS TO THE REVIEWERS

The authors acknowledge with gratitude the Reviewers’ effort in preparing the evaluation of the manuscript. The authors consider the suggestions and queries for having improved the manuscript significantly. The authors are very grateful to the Reviewers.

REVIEWER #1: 

The authors thank the Reviewer for these positive comments. We have rewritten the Introduction and, as suggested, incorporated the additional background information in other sections. Further, we have provided additional information on quality assessment in the Material and method section. The Result and Discussion sections have been abridged substantially, i.e., 17% and 22%, respectively. The authors have put a great deal of effort into shortening the sections without losing the sense of coherence. 

COMMENTS:

Q1: Line 40-44: The first sentence required 64 references, yet the second and, more informative sentence, remains unreferenced?

R1: The in-text references #1-64, comprising all the studies of the present review, were actually a technical feature facilitating a chronological order of the study material from the beginning. In the second sentence of the paragraph, three references have been added (Page 3, Line 42-44).

Q2: Sections “Classification of cutaneous burns”, “Contact burn injury method”, and “Evaluation of contact burn injury-induced changes” are all rather lengthy and feel disjointed between the introduction and the methods. Consider condensing these sections to incorporating them into the discussion, such that following the purpose of the review, the manuscript leads seamlessly into the methods.

R2: We agree that the subsections in the Introduction section could be incorporated elsewhere. We now have a single conjoined Introduction.

The rest of the information in the mentioned sections has been condensed and added in the Discussion.

Q3: With numerous acronyms, consider removing SHA. Numerous times throughout the manuscript, secondary hyperalgesia is mentioned but not short formed, while SHA is short formed. This feels rather odd when reading.

R3: The authors agree that it is inconsequential to switch between ‘secondary hyperalgesia area’ and ‘SHA’. However, since ‘SHA’ is mentioned numerous times (41 in the main text of the revised manuscript), we have changed ‘secondary hyperalgesia area’ to ‘SHA’ in all relevant places. There is, however, a distinction between ‘secondary hyperalgesia’ and ‘secondary hyperalgesia area’, and in order to improve readability, the authors have kept ‘secondary hyperalgesia’ unabbreviated.

Q4: Line 208-210. 73 records assessed for eligibility, 12 excluded, 5 added, to total 64 does not add up. (73-12+5=66)

R4: The authors thank the Reviewer for pointing out this error. We excluded 14 studies and not 12. This has been changed now.

Q5: The authors do not discuss their quality assessment until the limitations section of the review. Why are these not further discussed in greater detail throughout the manuscript? 

R5: Thank you for raising a relevant objection. Quality assessment using the Oxford Quality Scoring System can only be performed on RCTs. Therefore, quality assessment scores have only been made for the intervention studies and are thus only relevant when discussing these studies. However, we agree that more information could be provided about the quality scores. Consequently, the authors have put more effort into discussing quality scoring throughout the manuscript and have added additional information about quality scoring in the limitations sub-section.

Q6: The methods do not provide enough detail to interpret the quality outcomes (i.e. what does a score of 2 mean?). To that end, authors suggest over half of the intervention studies received a score of 0 on their chosen scale. What does a score of 0 represent? The authors have not provided information to interpret these values given in their tables. Additionally, this scale is more lenient than Cochrane’s suggested tool. Why were so many studies with quality such low quality included in the review and the discussion? The justification that “most were hypothesis driven pragmatic studies” does not accommodate for both methodological quality or author bias. This needs to be better addressed in the results and discussion.

R6: The authors thank the Reviewer for these suggestions. We have added additional information in the method section about the interpretation of the quality scores (please, also cf. R5 above).

During the revision of the manuscript, the authors became aware of an erroneous scoring procedure of the papers. The Oxford Quality Scoring System evaluates information about randomization, blinding, and subject dropouts. Studies get subtraction of a point if the randomization process and/or the blinding process was described and was inadequate. However, in our initial scoring of the studies, we subtracted a point if the information was missing. This resulted in a lot of studies receiving a score of 0 even though they were described as randomized, double-blind studies. We believe this may have given an inaccurate picture of the quality of the studies. The quality scores have, thus, been updated.

As mentioned in R5, we have updated the discussion regarding the quality scores and the inclusion of low-quality studies. We believe that valuable information on methodological aspects of the somatosensory testing would potentially have been lost by excluding intervention studies based on quality scores.

Q7: I’m curious as to why the authors did not perform any meta-analysis of findings? With the high number of included studies, it would have been beneficial for some of the more common outcomes to be included in separate meta analyses.

R7: In order to perform a meta-analysis, the authors would have had to look at a number of studies employing the same intervention and using identical somatosensory testing methodology. However, given the heterogeneity of the CBI methodologies applied, we did not find it meaningful to perform a meta-analysis.

Q8: Line 785 – what about capsaicin as well? The heat capsaicin model is very well documented.

R8: The Reviewer can rightfully claim that the authors could have mentioned the well documented heat-capsaicin model here. We have changed the sentence now.

Q9: Table 6 – These seem to be rather vague recommendations for future studies. Specifically, how long should stimulus be applied? To what temperature? These details would be expected in a summary table outlining the updated recommended methodology, as alluded to in the abstract.

R9: We agree with the Reviewer. The information in Table 4 has now been condensed, providing the most relevant methodological information available (Page 50-51).

REVIEWER #2

The authors thank the Reviewer for the positive comments. We have made an effort to abridge and streamline the manuscript further.

COMMENTS: 

Q1: Page 12, line 125: "QST…….by application of graded chemical, electrical,…." What chemical and electrical stimuli are the authors referring to? To the reviewer`s knowledge, classical QST protocols do not employ such stimuli.

R1: The authors acknowledge that ‘classic’ QST-assessments mainly involve thermal or mechanical stimulations. However, several studies, including some studies from the current review, employ electrical stimuli or chemical stimuli. See 'Response to Reviewers. Letter for references. 

Q2: I would like to commend the authors on their consistent use of terminology, as they are certainly well aware of the ongoing controversy regarding 'central sensitization' being used as a broad term in human studies in contrast to the very well-defined neuronal construct in basic animal studies. I am glad to see that the psychophysical terminology was used and distinctions between primary and secondary hyperalgesia were made appropriately. This was especially important for the interventional trials as some drugs will affect the primary input and the 2HA, respectively.

However, with respect to the "timing of intervention" it would be interesting to know if the studies aimed at investigating whether the drugs affected the induction of hyperalgesia or lead to a reversal of an already established state of hyperalgesia? In my personal opinion, only the former would constitute a true "anti-hyperalgesic effect". The latter could just be called "analgesic effect" considering CBI as a human surrogate pain model.

R2: Thank you very much for the relevant question. The authors agree that the timing of the intervention is an important factor potentially affecting the attenuation of hyperalgesia. The timing of interventions has indeed already been indicated for the individual studies in Table 3, as well as in the ‘Study outcomes – Intervention studies’ sub-section in the Results. 

Q3: In table 6 the use of "assessments of central mechanisms" is recommended. These tests are merely listed in the outcomes section in the introduction.

Please explain the methods/paradigms in a bit more detail, especially with regard to what mechanisms are potentially investigated with the respective paradigms.

R3: The assessments of central mechanisms refer predominantly to an evaluation of the development of secondary hyperalgesia and to a change in the response to temporal summation. Describing the specific QST outcomes is not within the scope of this review, and we refer to additional literature on this subject. However, the results regarding secondary hyperalgesia and temporal summation have been described throughout the manuscript. As mentioned in R9 (Reviewer #1) we have updated Table 6 (now Table 4) and condensed the information to only include the most important methodological information.

---

## [Decision Letter · Decision Letter 1]

11 May 2021

PONE-D-20-38717R1

Methodology and applicability of the human contact burn injury model: A systematic review

PLOS ONE

Dear Dr. Springborg,

Thank you for submitting your manuscript to PLOS ONE. After careful consideration, we feel that it has merit but does not fully meet PLOS ONE’s publication criteria as it currently stands. Therefore, we invite you to submit a revised version of the manuscript that addresses the points raised during the review process.

Review 2 has made a number of important suggestions, which the authors should consider. There is a need to consider the effect of heterogeneity between studies and the impact this has on the overall interpretation, as well as update to current standards. I am hopeful that, with a fair response to these comments, the manuscript will be satisfactory and ready for publication. 

We look forward to receiving your revised manuscript.

Kind regards,

John Kramer

Academic Editor

PLOS ONE

Journal Requirements:

Reviewers' comments:

Reviewer's Responses to Questions

**Comments to the Author**

1. If the authors have adequately addressed your comments raised in a previous round of review and you feel that this manuscript is now acceptable for publication, you may indicate that here to bypass the “Comments to the Author” section, enter your conflict of interest statement in the “Confidential to Editor” section, and submit your "Accept" recommendation.

Reviewer #1: All comments have been addressed

Reviewer #2: All comments have been addressed

2. Is the manuscript technically sound, and do the data support the conclusions?

Reviewer #1: Yes

Reviewer #2: Yes

3. Has the statistical analysis been performed appropriately and rigorously? 

Reviewer #1: N/A

Reviewer #2: Yes

4. Have the authors made all data underlying the findings in their manuscript fully available?

Reviewer #1: Yes

Reviewer #2: Yes

5. Is the manuscript presented in an intelligible fashion and written in standard English?

Reviewer #1: Yes

Reviewer #2: Yes

6. Review Comments to the Author

Reviewer #1: The authors have substantially improved the manuscript through the revision process. Many redundancies have been removed, improving the readability of the manuscript. The majority of my comments have been adequately addressed. I have a few minor comments for the authors consideration. The first should be very easy to include, and the remaining two may be worth consideration (or perhaps authors already considered this and sought not to include as it did not improve the interpretation).

PRISMA Diagram and checklist. These are the old versions of the PRISMA diagram and checklist. These were both updated in 2020. Consider updated to keep in line with PRISMA guidelines.

I previously suggested authors further discuss their quality assessment. Following revisions, the authors have pointed out that only intervention studies were assessed for quality. Are there no available tools to assess article quality for non-intervention studies? To that end, while the quality of intervention studies has been assessed, do the authors believe there would be any value in also using a tool to evaluate the quality of the data/findings from each study (using the GRADE tool perhaps)?

Throughout the results and discussion, there is clearly much heterogeneity in study findings. The general sense from the review is there is still “much work to do” in terms of using this model. An example of this would be the perplexing findings of the non-responsiveness of the model to anti-inflammatory drugs, despite being an inflammatory model. One avenue to address these concerns may be in weighing the quality of findings (in terms of Oxford and GRADE tools) to reach some more definitive conclusions. As it stands now, the quality assessment for the most part appears to be for the reader’s information, rather than directly used to help guide the synthesis of findings. It would be interesting to see if incorporating study quality alongside such contrasting findings could provide some clarity in such circumstances.

Reviewer #2: (No Response)

7. PLOS authors have the option to publish the peer review history of their article (what does this mean?). If published, this will include your full peer review and any attached files.

Reviewer #1: No

Reviewer #2: No

---

## [Author Response · Author response to Decision Letter 1]

10 Jun 2021

B. COMMENTS TO THE REVIEWERS

REVIEWER #1: 

The authors thank the reviewer for the kind comments and the help in improving the manuscript.

Q1: PRISMA Diagram and checklist. These are the old versions of the PRISMA diagram and checklist. These were both updated in 2020. Consider updated to keep in line with PRISMA guidelines.

R1: The authors acknowledge the reviewer for pointing this out. The updated versions of the PRISMA diagram and checklist from 2020 have been attached to the submission.

Q2: I previously suggested authors further discuss their quality assessment. Following revisions, the authors have pointed out that only intervention studies were assessed for quality. Are there no available tools to assess article quality for non-intervention studies? To that end, while the quality of intervention studies has been assessed, do the authors believe there would be any value in also using a tool to evaluate the quality of the data/findings from each study (using the GRADE tool perhaps)?

R2: There are several tools for assessing the quality of non-randomized studies. However, the authors have not been able to find a single tool that would be able to assess the quality of all the non-intervention studies in an intelligible fashion due to the considerable heterogeneity among these studies. Thus, the following sentence has been added to the Discussion:

- Page 48, Line 627-633: ‘Attempts were made at performing a quality scoring of the non-interventional studies. Incorporating study quality could provide clarity in studies presenting with contrasting results. Two potential scoring systems were considered: the GRADE tool [1,2] and the “checklist for the assessment of the methodological quality” [3]. However, due to considerable heterogeneity across the studies regarding methodology, outcome variables, and statistical processing, the authors decided to abandon the attempt to make a relevant quality scoring.’

Q3: Throughout the results and discussion, there is clearly much heterogeneity in study findings. The general sense from the review is there is still “much work to do” in terms of using this model. An example of this would be the perplexing findings of the non-responsiveness of the model to anti-inflammatory drugs, despite being an inflammatory model. One avenue to address these concerns may be in weighing the quality of findings (in terms of Oxford and GRADE tools) to reach some more definitive conclusions. As it stands now, the quality assessment for the most part appears to be for the reader’s information, rather than directly used to help guide the synthesis of findings. It would be interesting to see if incorporating study quality alongside such contrasting findings could provide some clarity in such circumstances.

R3: The authors agree that it might be interesting to review the different studies in light of their quality scores. However, there are still relatively few studies evaluating each drug group, and these studies use a very different methodology in terms of administration routes, outcome measures, etc. When reviewing each drug group, the general conclusion does not seem to change when only the high-quality studies are included (Oxford quality score 3-5). Thus, the authors have added the following sentence to the Discussion:

- Page 46-47, Page 590-597: ‘Interestingly, the studies evaluating anti-inflammatory drugs, i.e., NSAIDs and glucocorticoids, have mostly been negative but have received varying quality scores, raising the question of whether the results are correlated with the quality of the study. However, of the six studies evaluating NSAIDs, three different drugs and four different routes of administration were used along with several different outcome measures, obstructing a valid comparison between the studies. Further, among the three high-quality studies, only one study could be considered yielding a positive outcome [4], thus not changing the general conclusion of this review.’

REFERENCES

1. Atkins D, Best D, Briss PA, Eccles M, Falck-Ytter Y, Flottorp S, et al. Grading quality of evidence and strength of recommendations. BMJ. 2004;328(7454):1490. doi: 10.1136/bmj.328.7454.1490 [doi];328/7454/1490 [pii].

2. Guyatt GH, Oxman AD, Vist GE, Kunz R, Falck-Ytter Y, Alonso-Coello P, et al. GRADE: an emerging consensus on rating quality of evidence and strength of recommendations. BMJ. 2008;336(7650):924-6. doi: 336/7650/924 [pii];10.1136/bmj.39489.470347.AD [doi].

3. Downs SH, Black N. The feasibility of creating a checklist for the assessment of the methodological quality both of randomised and non-randomised studies of health care interventions. J Epidemiol Community Health. 1998;52(6):377-84. doi: 10.1136/jech.52.6.377. PubMed Central PMCID: PMCPMC1756728.

4. Stubhaug A, Romundstad L, Kaasa T, Breivik H. Methylprednisolone and ketorolac rapidly reduce hyperalgesia around a skin burn injury and increase pressure pain thresholds. Acta Anaesthesiol Scand. 2007;51(9):1138-46. doi: 10.1111/j.1399-6576.2007.01415.x.

---

## [Editor Report · Decision Letter 2]

6 Jul 2021

Methodology and applicability of the human contact burn injury model: A systematic review

PONE-D-20-38717R2

Dear Dr. Springborg,

We’re pleased to inform you that your manuscript has been judged scientifically suitable for publication and will be formally accepted for publication once it meets all outstanding technical requirements.

Kind regards,

John Kramer

Academic Editor

PLOS ONE

Additional Editor Comments (optional):

Unclear to me why Figure 1 (PRISMA diagram) is referenced in the introduction. Please consider removing the "?" in Table 4. 
---

## [Editor Report · Acceptance letter]

21 Jul 2021

PONE-D-20-38717R2 

Methodology and applicability of the human contact burn injury model: A systematic review 

Dear Dr. Springborg:

I'm pleased to inform you that your manuscript has been deemed suitable for publication in PLOS ONE. Congratulations! Your manuscript is now with our production department. 

Kind regards, 

on behalf of

Dr. John Kramer 

Academic Editor

PLOS ONE